# Loss of α2-6 sialylation promotes the transformation of synovial fibroblasts into a pro-inflammatory phenotype in arthritis

Yilin Wang[1], Aneesah Khan[1], Aristotelis Antonopoulos [2], Laura Bouché[2], Christopher D. Buckley [3,4,5], Andrew Filer [3,5], Karim Raza[3,6], Kun-Ping Li [7], Barbara Tolusso[5,8], Elisa Gremese[5,8], Mariola Kurowska-Stolarska [1,5], Stefano Alivernini [5,8,9], Anne Dell[2], Stuart M. Haslam [2] & Miguel A. Pineda [1,5✉]

In healthy joints, synovial fibroblasts (SFs) provide the microenvironment required to mediate homeostasis, but these cells adopt a pathological function in rheumatoid arthritis (RA). Carbohydrates (glycans) on cell surfaces are fundamental regulators of the interactions between stromal and immune cells, but little is known about the role of the SF glycome in joint inflammation. Here we study stromal guided pathophysiology by mapping SFs glycosylation pathways. Combining transcriptomic and glycomic analysis, we show that transformation of fibroblasts into pro-inflammatory cells is associated with glycan remodeling, a process that involves TNF-dependent inhibition of the glycosyltransferase ST6Gal1 and α2-6 sialylation. SF sialylation correlates with distinct functional subsets in murine experimental arthritis and remission stages in human RA. We propose that pro-inflammatory cytokines remodel the SF-glycome, converting the synovium into an under-sialylated and highly pro-inflammatory microenvironment. These results highlight the importance of glycosylation in stromal immunology and joint inflammation.

[1] Institute of Infection, Immunity and Inflammation, University of Glasgow, Glasgow, UK. [2] Department of Life Sciences, Imperial College London, London, UK. [3] Rheumatology Research Group, Institute for Inflammation and Ageing, College of Medical and Dental Sciences, University of Birmingham, Queen Elizabeth Hospital, Birmingham, UK. [4] The Kennedy Institute of Rheumatology, University of Oxford, Oxford, UK. [5] Research into Inflammatory Arthritis Centre Versus Arthritis (RACE), Glasgow, Birmingham, Newcastle Oxford, UK. [6] Sandwell and West Birmingham Hospitals NHS Trust, Birmingham, UK. [7] Institute of Chinese Medicinal Sciences, Guangdong Pharmaceutical University, Guangzhou, China. [8] Division of Rheumatology, Fondazione Policlinico Universitario A. Gemelli IRCCS, Rome, Italy. [9] Division of Rheumatology, Università Cattolica del Sacro Cuore, Rome, Italy. ✉email: miguel.pineda@glasgow.ac.uk

Rheumatoid arthritis (RA) is a chronic inflammatory condition of the joints affecting 0.3–1% of the world's population. RA has been historically described as an autoimmune condition, where the central component of disease pathogenesis relies on aberrant responses of immune cells leading to the destruction of bone and cartilage. Recent findings have now shown that RA pathophysiology merges autoreactive immunity with genetic, epigenetic, and environmental factors that are responsible for disease initiation. RA starts with a pre-clinical phase involving activation of immune mechanisms in the absence of clinical symptoms[1]. Later, auto-amplificatory loops recruit macrophages, T cells, and other immune cells to the joint. Local inflammation is perpetuated through cytokine networks dominated by TNF, IL-6, IL-1β, and chemokines such as Ccl2 or CXCL5[2,3]. Not surprisingly, biologic Disease-Modifying Anti-Rheumatic Drugs (bDMARS) inhibiting these cytokines are the treatments of choice in the clinic. Even though such gold standard treatments can substantially improve the life quality of thousands of patients[1,4], they can also induce serious adverse effects as a consequence of their immunosuppressive nature. Moreover, 30–40% of RA patients do not respond to them completely, suggesting that key events underpinning pathogenesis remain elusive. In fact, the cellular and molecular basis of why inflammation does not resolve in RA remains unanswered.

Synovial fibroblasts (SFs) are major components of the synovial membrane, a highly specialized mesenchymal tissue lining the joint cavity. In the synovium, two main microdomains can be described: the lining layer (directly exposed to the synovial space) and sub-lining layers. Due to their anatomical location, SFs provide the required nutritional and structural joint support and they were initially considered as cells lacking any substantial impact on immune function. However, SFs adopt a key immunopathological role in RA, responding to inflammatory cytokines and promoting tissue damage[2,3,5,6]. Thus, despite being cells of non-immune origin, SFs have a central role to perpetuate local immune responses in the synovial joint, delivering region-specific signals to infiltrating immune cells and contributing to bone and cartilage degradation[7,8]. Interestingly, Single Cell Transcriptomics have demonstrated that SFs comprise distinct functional subsets that correlate with their anatomical location and activation of pathological pathways[9,10], suggesting that SF-dependent immunity may be far more complex than anticipated. Because of their non-immune origin and their highly specialized function, interventions aiming at SFs—or specific SFs subsets—may modify disease progression without significant immunosuppression, although clinical targets have not yet been found.

Functional glycomics, an emerging discipline focused on defining the structures and functional roles of carbohydrates (glycans) in biological systems, could offer such fibroblast-specific molecular targets[11,12]. Some elegant studies have shown the potential impact of glycan regulation in multiple aspects of RA pathophysiology. Reduced sialylation of N-glycans is a feature of pathogenic immunoglobulins in RA patients and their glycosylation profile shows predictive potential for disease progression[13–17]. Galectins, a family of proteins that bind to galactose-containing glycans, are key modulators of synovial inflammation[18–20]. However, little is known about the glycosylation profile of SFs, or whether or not this varies in health and disease. Glycosylation modulates cellular interactions and responses to immunomodulatory carbohydrate-binding proteins. In fact, altered glycosylation is a hallmark of chronic inflammatory conditions. In cancer, cytokines induce changes in the cell glycome leading to local inflammation via control of cell adhesion, migration, and signal transduction[21–23]; mechanisms that are also associated with the migration and pathogenicity of SFs in RA. Furthermore, galectin-3 is upregulated in RA[24], and galectin-3[−/−] mice show reduced pathology in experimental arthritis[20]. Moreover, exogenous galectin-3 significantly upregulates CCL2, CCL3, and CCL5 in synovial but not in dermal fibroblasts[19], suggesting that the synovial microenvironment can induce tissue-specific glycosylation in the stromal compartment.

We hypothesized that the cytokine milieu in the inflamed joint controls distinct SF-glycosylation, which in turn, regulates cell recruitment and inflammatory responses. We investigate changes in the SF glycome that could be related to their inflammatory activity. By combining transcriptomic and glycomic analysis, we report that the transformation of SFs into pro-inflammatory cells in experimental arthritis is associated with glycan remodeling, which involves the reduction of terminal sialylation in Thy1 (CD90)[+] sub-lining SFs upon TNF stimulation. Notably, enzymatic removal of sialic acid is sufficient to induce inflammatory pathways in the absence of further stimulation. We also show that low sialylation of SFs is associated with disease remission in human RA, supporting the idea that the stromal glycome could be used for the development of disease biomarkers or therapeutics.

## Results

### Anatomical location and inflammation shape fibroblast glycosylation.
First, we decided to test whether glycosylation could be conditioned by local microenvironments. We expanded fibroblasts from RA joint replacement surgery and matched dermal fibroblasts, as a reflection of non-inflammatory environments. We also isolated SFs from osteoarthritis (OA), as an example of a less inflammatory joint disease than RA. General glycosylation was evaluated using lectin-binding assays by immunofluorescence (Supplementary Fig. 1a) and flow cytometry (Supplementary Fig. 1b, c). All fibroblasts bound most of the tested lectins, confirming the presence of a rich and diverse glycocalyx. Overall, human SF-glycome seemed to be rich in galactose-containing glycans (RCA[+], ECA[+]) containing Poly-LacNAc extensions (LEL[+]) and α1–6 core-fucosylated N-glycans (AAL[+]), in contrast to the lack of α1,2 fucosylation on glycan antennae (UEA[−]). We also observed significant differences between anatomical locations (dermal vs synovium: PNA, Jacalin and ECA binding) and well as between distinct inflammatory conditions (RA vs OA, LEL, and ECA binding) (Supplementary Fig. 1), supporting the potential link between local inflammatory mediators and glycan remodeling.

Next, we gathered further evidence using the available RNA-Seq data set generated by Slowikowski et al.[25], where human SFs were stimulated with TNF (1 ng/ml) and IL-17 (10 ng/ml). Glycosyltransferases and glycosidases are the enzymes synthesizing the cell glycome, and so we evaluated the relative expression of these enzymes in Slowikowski's data set. TNF significantly modulated the biosynthetic pathways of branched glycans, like GCNT2, along with the upregulation of α2-3-sialyltransferases (ST3Gal1, ST3Gal2, and ST3Gal4) (Supplementary Fig. 2a). This suggested that TNF may directly modify SF-glycosylation. On the other hand, no effect was observed in response to IL-17 (Supplementary Fig. 2b), indicating that changes in glycosylation are linked to distinct cytokines. Nonetheless, results shown in Supplementary Figs. 1 and 2a, b should be analyzed with caution. Cells were isolated from patients undergoing joint replacement surgery, that have been exposed to long-term immunomodulatory treatments affecting the cytokine–glycosylation axis that we intend to study. Thus, to further assess our hypothesis, naive SFs were expanded from the synovium of healthy mice. Murine SFs were stimulated with a panel of regulatory and pro-inflammatory factors and subsequently stained with PHA, a lectin that binds complex branched N-glycans. We used IL-22 as

an example of a cytokine with both pro- and anti-inflammatory activity in RA that specifically targets stromal cells[26]. Corroborating the link between distinct glycosylation and effector responses, the combination of inflammatory factors with IL-22 exerted diverse effects on the expression of branched glycans (Supplementary Fig. 2c). Changes in PHA-binding in response to immunomodulators might be reflective of SFs activation as glycans recognized by PHA were detected in joint areas with a strong SF-mediated inflammation (Supplementary Fig. 2d).

**RNA-Seq identifies glycosylation pathways associated with SF pathogenesis**. Overall, our preliminary results in mouse and human cells supported the hypothesis that immune mediators found in local microenvironments control SF glycosylation. This suggested that glycan expression and inflammation could be intertwined events and we wanted to describe the mechanisms in detail. However, SFs isolated from joint arthroplasty do not fully represent disease pathophysiology, as cytokine–glycosylation pathways could be significantly affected by long-term immuno-suppressive treatments. Therefore, although these results cannot provide conclusive mechanistic data, they offered the required scientific support to utilize animal models, a system that is more suitable to investigate basic physiological pathways. We chose the murine model of Collagen-induced arthritis (CIA) because it shares hallmarks of human disease of high relevance to this study, such as the hyperplasia of the synovial membrane, inflammatory infiltration of the synovium, and pannus formation. SFs from healthy and CIA mice (Clinical scores >8) were sorted by flow cytometry [CD45−CD31−Podoplanin+] (Fig. 1a) and RNA was immediately purified for transcriptome sequencing (RNA-Seq) analysis upon polyA selection. We recovered a greater number of SFs from CIA joints compared to healthy joints, with elevated expression of podoplanin (Fig. 1a), reflecting their hyperplasia and activation reminiscent of human RA. Consistently, Principal Component Analysis of the transcriptomic data confirmed the distinction between both groups (Fig. 1b). A list of differentially expressed (DE) genes in CIA SFs was generated including 298 upregulated genes and 88 downregulated [>2-fold, $p_{adj} < 0.01$, (Fig. 1b and Supplementary Fig. 3a)]. KEGG pathway enrichment analysis showed that "Rheumatoid Arthritis" as a disease pathway was significantly upregulated in CIA SFs (Supplementary Fig. 3b), validating the model. String Protein–Protein Interaction Networks Functional Enrichment Analysis[27] was applied to DE genes in CIA compared to healthy SFs, identifying 2 main functional networks: (i) cell cycle and cell division and (ii) inflammatory response (Fig. 1c), further demonstrating SF immune activation and hyperproliferation. Interestingly, GO-term analysis revealed that most of the proteins identified in the inflammatory network were glycoproteins and/or regulators of cell communication (Fig. 1c), suggesting a potential role for glycosylation in SF activation.

To fully dissect the transcriptomic profile of SFs glycosylation, we evaluated the differential expression of genes involved in different steps of glycosylation biosynthesis (glycosyltransferases and glycosidases involved in mannosylation, glycan chain branching and elongation, fucosylation, sialylation, and glycan degradation, Fig. 1d). Unsupervised clustering based on these glycan biosynthetic pathways separated naive and arthritic SFs in all cases (Fig. 1e), indicating that pro-inflammatory SFs are also defined by a characteristic glycosylation capability. The observed downregulation of enzymes of medial Golgi-branching N-acetylglucosaminyltransferases II, IV, and V (encoded by *Mgat2*, *Mgat4*, and *Mgat5*) in CIA SFs, along with the upregulation of β-1,4-Galactosyltransferase genes (*B4galt*) and *GCNT2* (Fig. 2e) suggested that these differences could modify

the extension and branching of antennae of N-glycans or the number of poly N-acetyllactosamine (linear repeats of Galβ1,4GlcNAcβ1,3) synthesized by β1,3N-acetylglucosaminyl-transferases. Likewise, terminal modifications of such structures may have reduced fucosylation or sialylation, given that fucosyltransferases *Fut10*, *Fut11*, and the sialyltransferases *St6gal1*, *St3gal2*, and *St3gal6* are significantly downregulated in CIA SFs (Fig. 1e).

**Sialylation is reduced in pro-inflammatory SFs**. Transcriptomics proved to be a powerful tool to delineate potential changes in cell glycosylation. However, unlike proteins or nucleic acids, glycans are not assembled in a template-driven process. Rather, glycosylation in the endoplasmic reticulum and Golgi is the result of combined actions of glycosyltransferases and gly-cosidases (Fig. 1d). Consequently, the prediction of structures based on transcriptomic data does not necessarily correlate with the final glycosylation profile, and further structural information is needed to generate reliable glycan structural conclusions. We used mass spectrometry (MS) based glycomics to define the N-glycome of murine SFs (Supplementary Fig. 4). N-glycans were isolated from cultured SFs, permethylated, and subjected to MS analysis. Annotation of MS peaks with most likely glycan structures was based on molecular ion composition, knowledge of biosynthetic pathways, and with the assistance of the bioinformatic tool glycoworkbench[28]. Most structures were annotated as high-mannose glycans or complex glycans, either core-fucosylated or non-fucosylated. Sequential addition of *N*-acetyllactosamines (LacNAcs) defined larger glycans, suggesting the presence of extended antennae and multi-branched structures. Nevertheless, shorter bi-antennary glycans constituted the most abundant type. Sialylation (including N-acetylneuraminic [Neu5Ac] and *N*-glycolylneuraminic acid [Neu5Gc]) was the most abundant capping modification, followed by α-galactosylation (Gal-αGal). Neither Gal-αGal, nor Neu5Gc is biosynthesized by humans. Because of the implications for potential translational work, the N-glycome of human OA SFs was defined as for the mouse cells (Supplementary Fig. 5). Like the mouse glycome, human SFs express a glycome dominated by high-mannose and short bi-antennary glycans. A full comparison of the main structural groups in mouse and human was conducted (Supplementary Fig. 6). The only significant difference between mouse and human structures was found in the expression of complex N-glycans containing a greater number of polyLacNAc groups in the human samples. This probably reflects the lack of α-galactosylation, which would prevent the addition of polyLacNAc groups. Nevertheless, these complex glycans represent <10% of both glycomes, indicating that human and mouse SF N-glycomes are well conserved. Importantly, the total amount of sialylated and fucosylated forms was comparable in both species.

Next, to identify specific glycan changes that could contribute to SF activation, the relative expression of individual N-glycans structures in healthy and CIA SFs were compared. Structures whose relative expression was lower than 0.02% of the total were excluded since the low expressed forms could easily add artefactual results. This data set containing 43 N-glycans were subjected to unsupervised hierarchical clustering to uncover expression patterns characteristic of inflammatory CIA SFs (Fig. 2). Interestingly, N-glycans were clustered into 6 discrete groups containing similar structural features: (i) *cluster 1987*: high-mannose, (ii) *cluster 2285*: LacNAc extended, (iii) *cluster 3026*: sialylated-fucosylated, (iv) *cluster 3271*: sialylated-LacNAc (v) *cluster 3258*: sialylated-LacNAc-fucosylated, and (vi) *cluster 2069*: simple non-sialylated. Three of these clusters contained

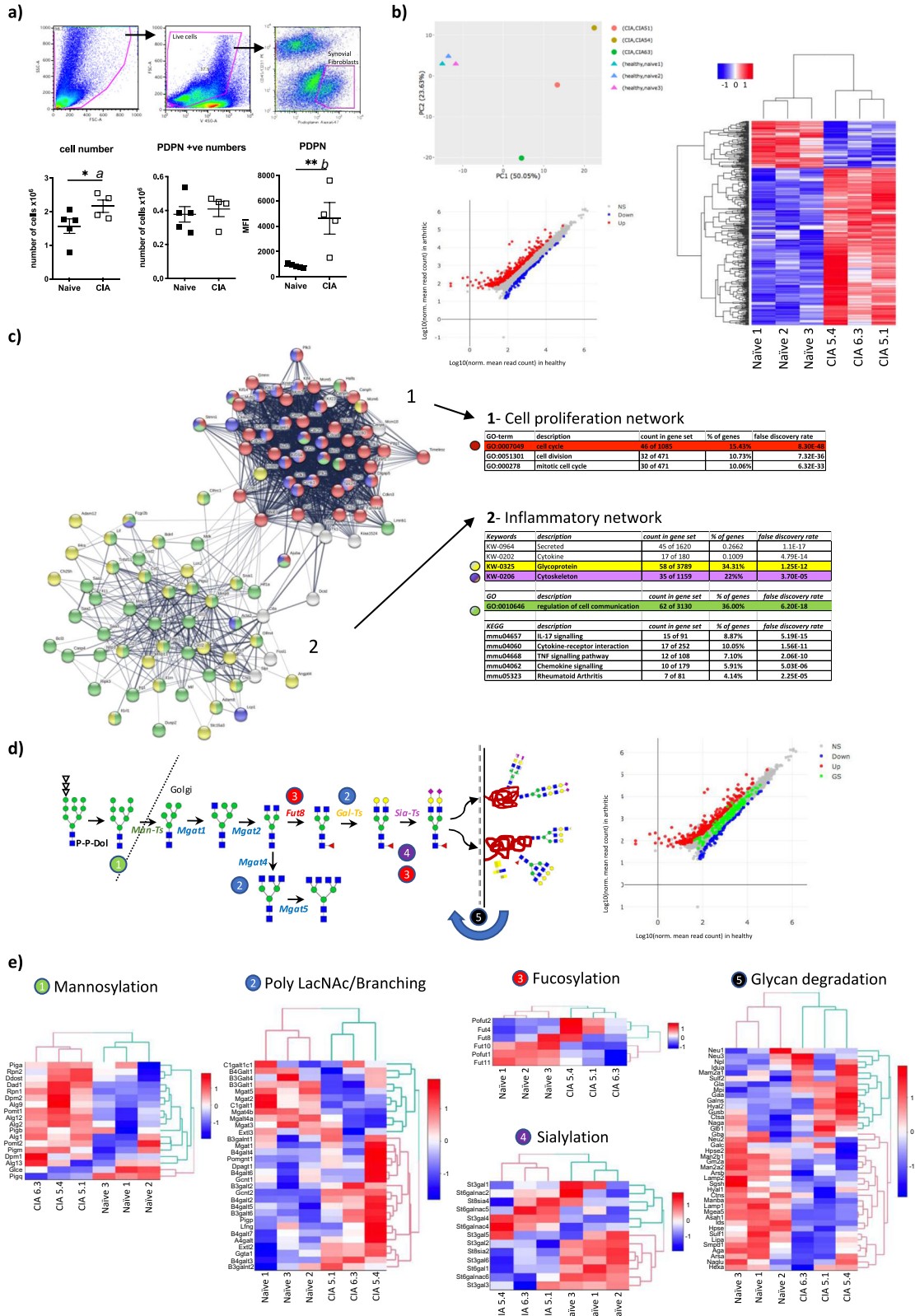

glycans that were significantly downregulated in CIA SFs: clusters 3026, 3271, and 3258, which had one characteristic in common: a significantly high proportion of sialic acid-containing glycans. In fact, 96% of all the sialylated glycans analyzed were found within these three clusters, strongly suggesting that reduced sialylation is associated with activated CIA SFs. The relative increase of related non-sialylated cores ($m/z$ 2244, 2069, 2110, and 2285) supports

the loss of sialylation. We also determined the murine O-glycome of healthy (Fig. 3a) and CIA SFs (Fig. 3b). O-glycans were isolated by reductive elimination, permethylated, and subjected to MS. SFs express core 1 and core 2 O-glycans, with limited LacNAc extensions. We identified multiple sialylated structures, but no fucosylation at all, in agreement with the N-glycome description. Likewise, relative expression of sialylated O-glycans

**Fig. 1 Inflammatory synovial fibroblasts have a distinct transcriptomic profile of glycosylation pathways.** Paws from healthy and arthritic mice were dissected and soft tissue was digested. **a** Live synovial fibroblasts (Zombie Violet⁻, Podoplanin⁺, CD45⁻, CD31⁻) were sorted by flow cytometry as shown in dot plots. The number of isolated cells, PDPN⁺, and expression of PDPN are shown. Data are presented as mean ± SEM. Naive n = 5, arthritic n = 4. Statistical significance was calculated using a one-tail unpaired t-test, *p < 0.05 and **p < 0.01. Actual p-values: a 0.0419, b 0.0055. **b** RNA was isolated (RIN > 9) from healthy (n = 3) and arthritic CIA (n = 3, scores of 9, 10, and 11) mice and subjected to bulk RNA-Seq (75 bp paired-end, 30 M reads). Principal component analysis (PCA) and differential expression (DE) of genes are shown. All detected genes are plotted as a scatter plot where x = gene expression healthy, y = gene expression arthritic. Genes that pass a threshold of $p_{adj}$ < 0.01 and |log2foldChange| > 2 in DE analysis are colored in blue when they are downregulated and red when they are upregulated in the arthritic (CIA) mice. Heatmap shows up and downregulated genes; [unsupervised clustering in rows and columns based on Euclidean distances]. **c** Function enrichment and network analysis regulated by synovial inflammation. STRING protein–protein interaction network (https://string-db.org) was performed on DE genes from **b**. Significantly modulated pathways and cellular components associated with DE genes in arthritic mice are shown in tables. [PPI enrichment p-value: < 1.0e−16]. K-means method gave two main functionally related clusters of genes, designated as "cell proliferation" and "inflammatory" cluster. Color code for nodes is, red: cell cycle, purple: cytoskeleton, yellow: glycoprotein, green: regulation of cell communication. **d** Analysis of glycogenes and glycan biosynthesis pathways. The illustration shows the N-glycan synthesis pathway. Scatter blot: as in **b** but with genes annotated in the glycan biosynthesis pathways highlighted in green. **e** Heatmaps of scaled and centered log₁₀ transformed normalized read counts showing only differentially expressed genes involved in glycosylation pathways.

was reduced in CIA SFs (Fig. 3c), corroborating the lower sialylation signature observed in N-glycans. Moreover, neither N-glycans, nor O-glycans showed any difference in the relative expression of Neu5Ac vs Neu5Gc.

**SFs from arthritic mice exhibit a decrease in α2-6-linked sialic acid.** Comparative glycomic analysis indicated that experimental arthritis was strongly associated with a downregulation in sialic-acid-containing glycans in SFs (Figs. 2 and 3). A retrospective examination of the transcriptomic data reveals a significant downregulation of St3gal6, St3gal2, and St6gal1 [fold-increase 0.73, 0.64, and 0.60 respectively] (Fig. 1e), supporting the reduced presence of sialylated glycans observed by MS (Figs. 2 and 3). However, it also showed a significant upregulation of St3gal4 mRNA [1.49 fold-increase (Fig. 1e)]. These apparently conflicting results could be explained by the differential regulation of sialic acid linkages that would not be detected in our MS-based studies. Two glycosidic bonds could be found in SFs: sialic acid-α2-3Gal and sialic acid-α2-6Gal, synthesized by six ST3 beta-galactoside alpha-2,3-sialyltransferases (ST3Gal1-6) and two ST6 beta-galactoside alpha-2,6-sialyltransferases (ST6Gal1-2), respectively. We used SNA and MAAII, lectins that specifically recognize sialic acid in α2-6 and α2-3-linkages. SNA binding was reduced in CIA SFs compared to naive SFs, whereas MAAII binding was not affected (Fig. 4a). These results indicate that the differential sialylation profile observed in CIA SFs is due to a specific reduction in α2-6-sialylation, presumably because of lower ST6-sialyltransferases expression. Furthermore, binding of galactose-recognizing lectins such as PNA and SBA was upregulated in arthritic SFs (Fig. 4a), probably reflecting increased terminal galactose in the under-sialylated glycome. Since inflammation does not change α2-3-sialylation of SFs, a reduced ratio of α2,6/α2,3-linked sialic acid might constitute a hallmark of inflammatory SFs. Supporting these findings, synovial membranes of healthy mouse joints had an α2-6 > α2-3-sialylation profile, as observed by immunofluorescence with SNA and MAAII, whilst inflamed joints in CIA mice show comparable levels of both sialic acid linkages (Fig. 4b).

**TNF down-regulates St6gal1 expression and α2-6-sialylation.** Transcriptomics, MS-based glycomics, and lectin-binding experiments concluded that reduced α2-6-sialylation is associated with inflammatory SFs. Next, we sought to identify the molecular mechanisms causing this shift. Several sialyltransferases (Sia-Ts) were expressed in SFs (Fig. 5a), including ST3Gal enzymes [St3gal1 >> St3gal2 > St3gal4 > St3gal3] and only one involved in α2-6-Sialylation, St6gal1. Enzymes involved in polysialic synthesis

(ST8Sia-Ts) were only marginally expressed, in agreement with MS glycomics data (Supplementary Fig. 4). We also evaluated the expression of enzymes involved in CMP-Neu5Ac synthesis (Cmas, Gne, Nanp, Nans, Slc35a1, Fig. 5a). CMP-NeuAc is the cytosolic donor for sialic acid, and a decrease in its intracellular levels could also lead to hyposialylated glycomes, regardless of Sia-Ts activity. However, these enzymes were highly expressed in both healthy and CIA SFs (Fig. 5a), suggesting that intracellular availability of sialic acid might not restrain the biosynthesis of sialosides. However, and rather unexpectedly, neither RNA-Seq analysis nor qPCR approaches detected expression of N-Acetylneuraminic Acid Phosphatase (Nanp), an enzyme that dephosphorylates sialic acid 9-phosphate to free sialic acid. Perhaps alternative bio-synthesis pathways operate in SFs, as in recent observations in CHO cells[29]. Cmah, an enzyme that converts CMP-Neu5Ac to CMP-Neu5Gc in murine cells, and Casd1 and Siae, genes that potentially modulate acetylation of sialic acid, were also detected, although expression was similar between healthy and CIA SFs.

To identify the mechanisms modulating sialylation, we used qPCR to evaluate the expression of selected genes (as in Fig. 5a) in response to IL-1β, IL-17, and TNF, key pro-inflammatory cytokines in RA. Confirming cell activation, all cytokines upregulated Il6 mRNA upon cytokine stimulation (Fig. 5b). No significant effect was seen in response to IL-17, in line with observations in Supplementary Fig. 2b. No difference was observed in Gne, Nans, Cmas, or SLC35A1 expression, corroborating that sialic acid biosynthesis is not the cause for the reduced sialylation observed in CIA. IL-1β increased expression of St6gal1 (in naive SFs) and St3gal4 (in CIA SFs), with an approximately 2-fold-increase. However, these changes were only mild compared with the significant downregulation (8-fold) of St6gal1 in response to TNF in both naive and CIA SFs (Fig. 5b). St6gal2 was not detected in mouse fibroblasts, suggesting that the TNF-ST6Gal1 link has a key role in the reduction of α2-6-sialylation during CIA. In consonance with additional qPCR results, SNA-binding was reduced only by TNF stimulation, but not by IL1β or IL-17 (Fig. 5c). TNF also downregulated ST6GAL1 mRNA in human SFs isolated from OA patients (Fig. 5d). Unlike the case of murine SFs, we observed the expression of ST6GAL2 in human cells, although its expression was lower than that observed for ST6GAL1. TNF also downregulated ST6GAL2 mRNA (Fig. 5d), corroborating the role of this cytokine in fibroblast desialylation in the human system. Finally, N-glycans from murine cells were isolated after TNF stimulation to conduct MS-based glycomic analysis. 12 out of 16 sialylated N-glycans showed a reduced expression (Fig. 5e), providing additional support to our findings.

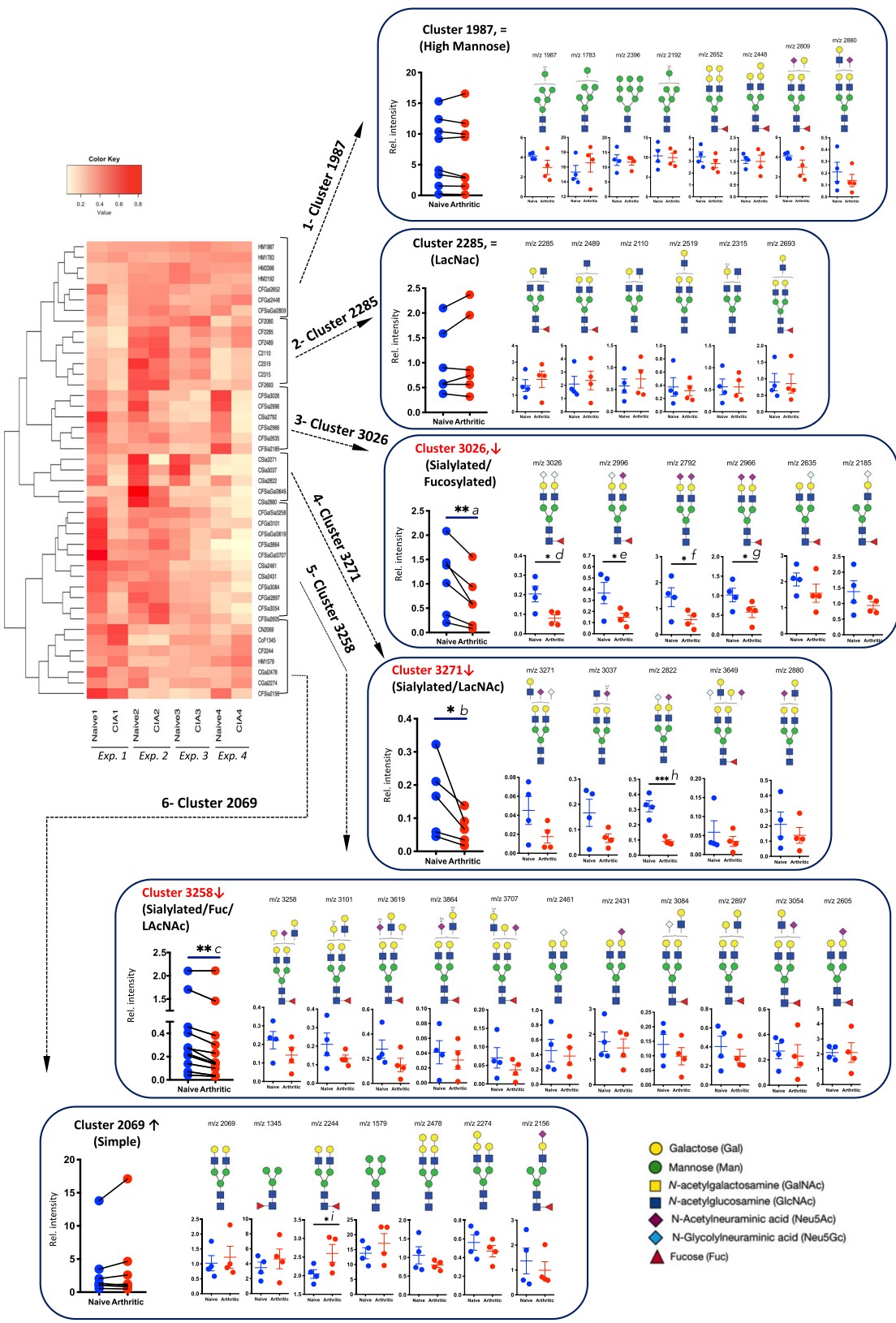

**Synovial fibroblast subsets are differentially sialylated**. The discovery of individual subsets of SFs has revolutionized our understanding of fibroblast biology in RA. Single-cell transcriptomic experiments have shown that fibroblast subsets localize to specific regions in the synovium contributing to different aspects of disease pathogenesis[9,10]. For example, CD90[+]FAPα[+] fibroblasts located in the synovial sub-lining are essential for the perpetuation of the inflammatory response, whereas CD90[−]FAPα[+] in the lining membrane is responsible for bone and cartilage damage[9]. Thereby, understanding which functional SF subset(s) lose sialic

**Fig. 2 Synovial fibroblasts from arthritic mice have a reduction in sialylated N-glycans.** N-glycans from synovial fibroblasts expanded from healthy or arthritic synovial fibroblasts were isolated and permethylated prior to MALDI-TOF MS analysis. 43 structures were selected (relative expression >0.02%) and MS peak area was quantified and normalized against total measured intensities. Unsupervised hierarchical clustering with Euclidean distance grouped structures into six main clusters. Structures present in each cluster are shown. Blue and red dots represent SFs isolated from healthy and CIA mice respectively. Data show the results from 4 independent experiments, where healthy and arthritic fibroblasts samples were processed in parallel. Before-after graphs show all structures present in the indicated clusters, where individual dots are the mean of relative expression from 4 experiments and similar structures in healthy and CIA cells are connected with lines. Statistical significance between healthy and CIA was assessed by one-tail paired $t$-test, *$p <$ 0.05 and **$p < 0.01$. Actual $p$-values: a 0.0046, b 0.0371, c 0.024. Relative expression for each glycan structure (shown with their $m/z$ value) is also shown in scatter plots showing mean and SEM, where each dot represents the relative expression of one independent experiment, $n = 4$, *$p < 0.05$ and **$p < 0.01$ using unpaired one-tail $t$-test. Actual $p$-values: d 0.0153, e 0.0385, f 0.0372, g 0.0478, h 0.0005, i 0.0420.

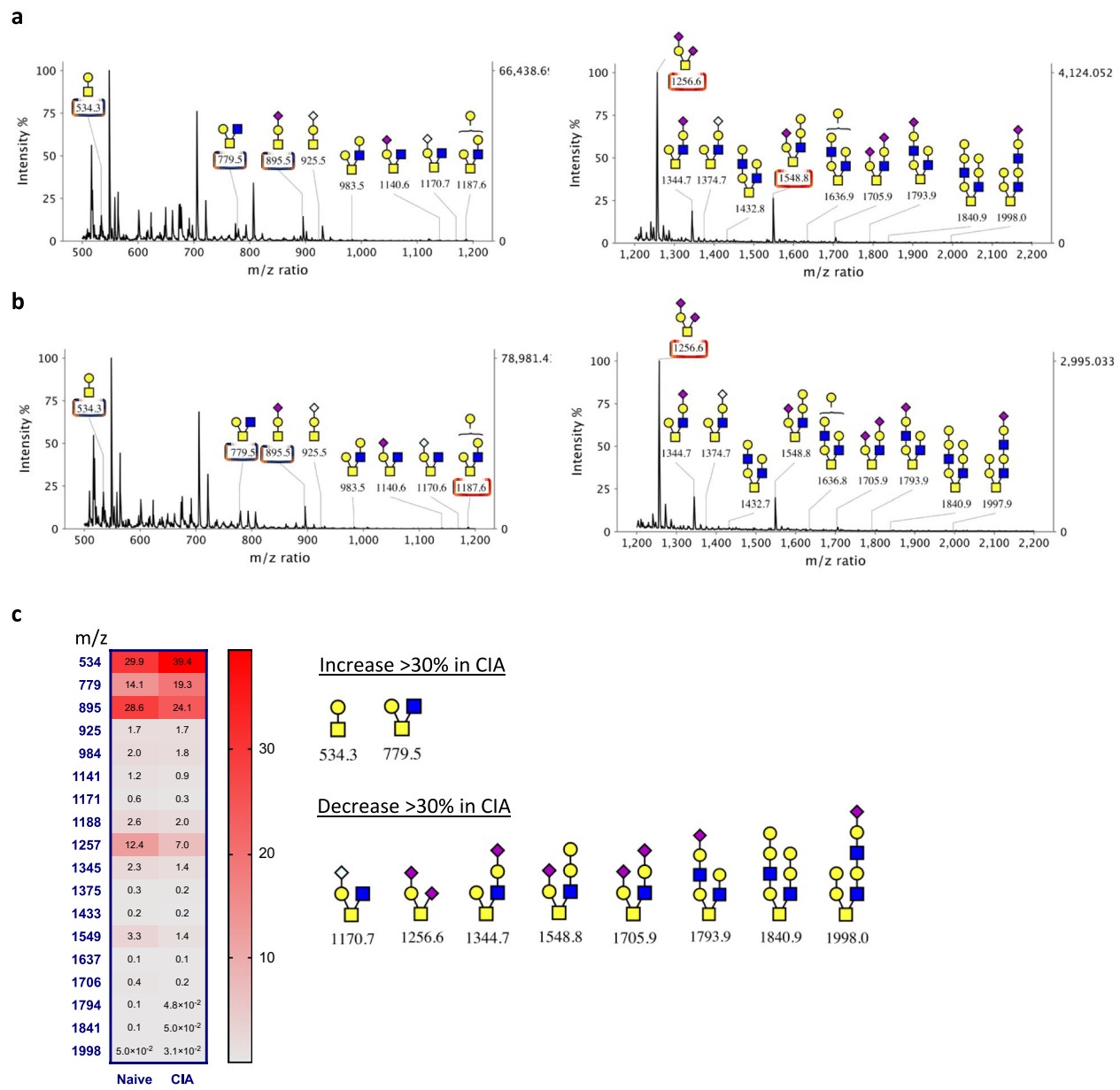

**Fig. 3 MALDI-TOF MS profiles of the permethylated O-linked glycans derived from murine SFs. a**, **b** Synovial fibroblasts were expanded from healthy or arthritic mouse synovium. O-glycans were isolated by reductive elimination and permethylated prior to MALDI-TOF MS analysis. The analysis was performed on merged SFs from **a** naive ($n = 3$) and **b** CIA ($n = 3$) mice. Blue boxes show the structures constituting 75% of all the glycome and red boxes show those completing 90% of total expression. **c** Heatmap showing the relative expression of each annotated form. Structures showing a differential expression (±30%) in the CIA mouse are shown.

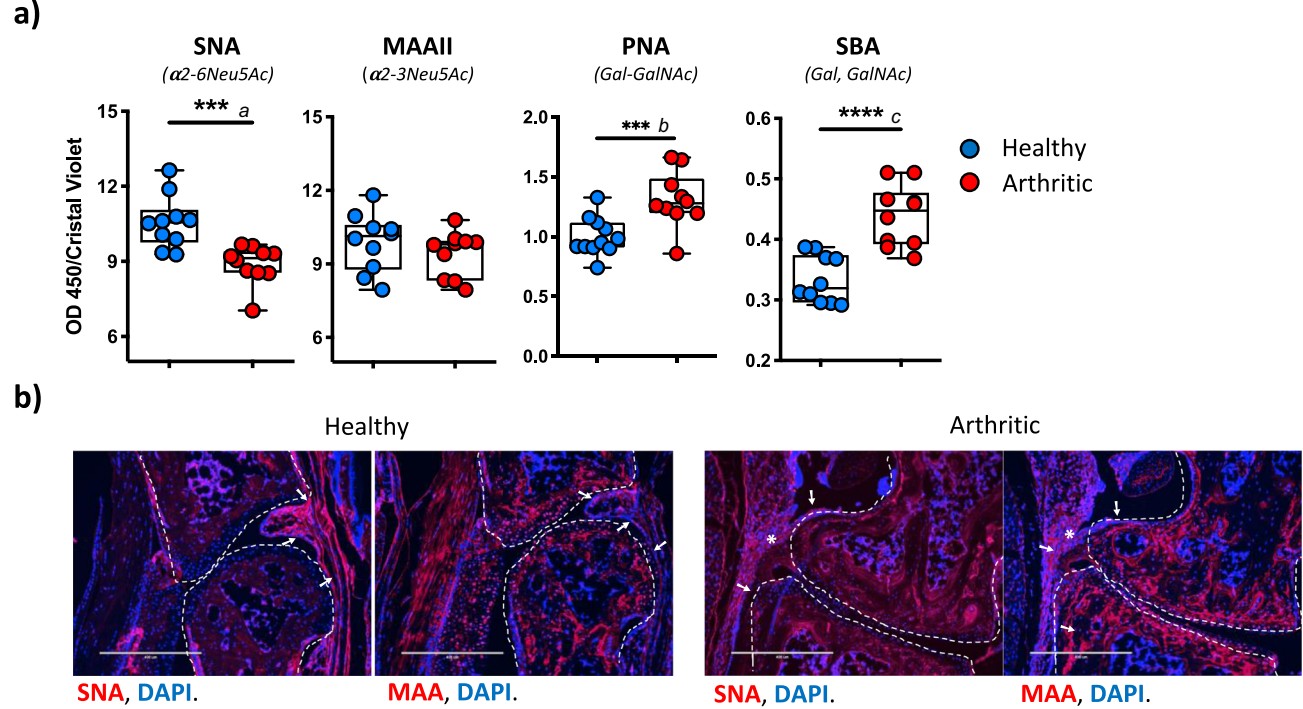

**Fig. 4 Synovial fibroblasts from arthritic mice have reduced levels of α2-6 but not α2-3-sialylation. a** SNA, MAAII, PNA, and SBA lectins were used in enzyme-linked lectin assay (ELLA) to evaluate the presence of α2-6 Sialic acid, α2-3 sialic acid, non-sialylated T antigen, and terminal N-acetylgalactosamine/galactose respectively. $n = 10$ biological replicates from three independent experiments. Statistics: two-tail unpaired $t$-test, $n = 10$. *$p < 0.05$, **$p < 0.01$, ***$p < 0.001$. Actual $p$-values: a 0.0007, b <0.00001, c 0.0018. Boxplots show 25th to 75th percentiles and median, whiskers from minimum to maximum. **b** Mouse joint sections were stained with biotinylated SNA and MAAII and fluorescently labeled streptavidin. Magnification ×20, scale bars: 400 μm. Lectin binding is shown in red, nuclei counterstain with DAPI in blue. Superimposed white elements indicate as follows; [dotted lines]: bone limits, [arrows]: synovial membrane, [asterisks]: local inflammation and infiltrating cells. The experiment was repeated twice with similar results.

acid during inflammation would establish the functional relevance of sialylation in SF-mediated pathogenesis. SNA and MAAII lectins were used to detect α2-6 and α2-3 sialic acid in CD90⁺ (sub-lining) and CD90⁻ (lining) SFs directly isolated from digested mouse synovium. PNA was used as a control, as sialic acid prevents its binding. SFs from healthy joints showed a higher affinity for SNA than MAAII, supporting the homeostatic role of α2-6-sialylation. Interestingly, naive unstimulated CD90⁺ SFs had a higher basal α2-6-sialylation in comparison to CD90⁻ SFs (Fig. 6a), perhaps suggesting that sialic acid content is related to functional and geographical SF specialization. Next, we FACS-sorted CD90⁺ and CD90⁻ SFs from naive and CIA joints to evaluate *St6gal1*, *Il6*, and *Mmp3* mRNA levels (Fig. 6b). Corroborating our results, *ST6Gal1* was found to be downregulated in CIA SFs, but only in CD90⁺ SFs. This agrees with recent reports showing that CD90⁺ sub-lining SFs are the main drivers of disease in RA synovial immune responses[9,10,30]. As expected, *Mmp3* and *Il6* were upregulated during arthritis and preferentially expressed by CD90- SFs, consistent with findings from Croft et al.[9]. In addition to *St6gal1*, SFs subsets also showed differential *Mmp3* and *Il6* expression (Fig. 6b), connecting function, geographical location, and glycosylation signature.

**α2-6-sialylation is associated with disease remission stages**. Given the downregulation of α2-6-sialosides in SFs during CIA, we postulated that a low sialylated SF-glycome would be a characteristic signature of strong inflammatory environments, especially those with high TNF concentration. To test this hypothesis, we compared the N-glycome of SFs expanded from CIA animals, separating them into cells from non-affected paws (low scores, LS-SFs) and very inflamed paws (high scores, HS-

SFs), taking advantage of the CIA model being asymmetrical. In addition, we included SFs from naive non-arthritic mice. Unsupervised clustering of N-glycan relative expression (assessed by MS) revealed one overexpressed cluster in LS-SFs (*cluster 3619*, Fig. 7a) containing 84% of sialylated structures. By contrast, *cluster 2080* was under-represented and it included almost exclusively non-sialylated glycans (Fig. 7a). Because α2-6-linked sialic acid is a blocker for galectin binding[31,32], low sialylated glycomes could be more sensitive to interactions with galectins, due to a higher exposure of terminal galactosides. This could be the case for LS SFs. To test this, naive, LS- and HS-SFs (as in Fig. 7a) were stimulated with recombinant galectin-3, described as a pro-inflammatory factor in RA[33]. IL-17 and TNF were used as inflammatory mediators that do not bind glycans. In all cases, IL-6 production confirmed SF activation. Indeed, all SF groups showed some ability to respond to IL-17 and TNF, being their response in direct correlation to their inflammatory status (Fig. 7b, c). However, LS-SFs did not respond to galectin-3, in clear contrast to HS-SFs (Fig. 7c) which were easily activated. This could be explained by the protective coating of sialic acid in the LS-SFs-glycome compared to that of the HS-SFs, preventing galectin-3 binding and consequent inflammatory response. There could also be differences in N-glycan branching amongst groups, which may affect galectin-3 binding. Nevertheless, LS-SFs still expressed more IL-6 than naive SFs, both in resting and stimulated cells. Thus, we cannot rule out that highly sialylated LS-SFs are in a pre-clinical phase transitioning towards more inflammatory conditions, rather than in an anti-inflammatory or remission stage.

Hence, to assess whether high content of sialic acid is a protective marker, we used SFs expanded from either untreated early RA patients (<12 months from joint symptoms beginning),

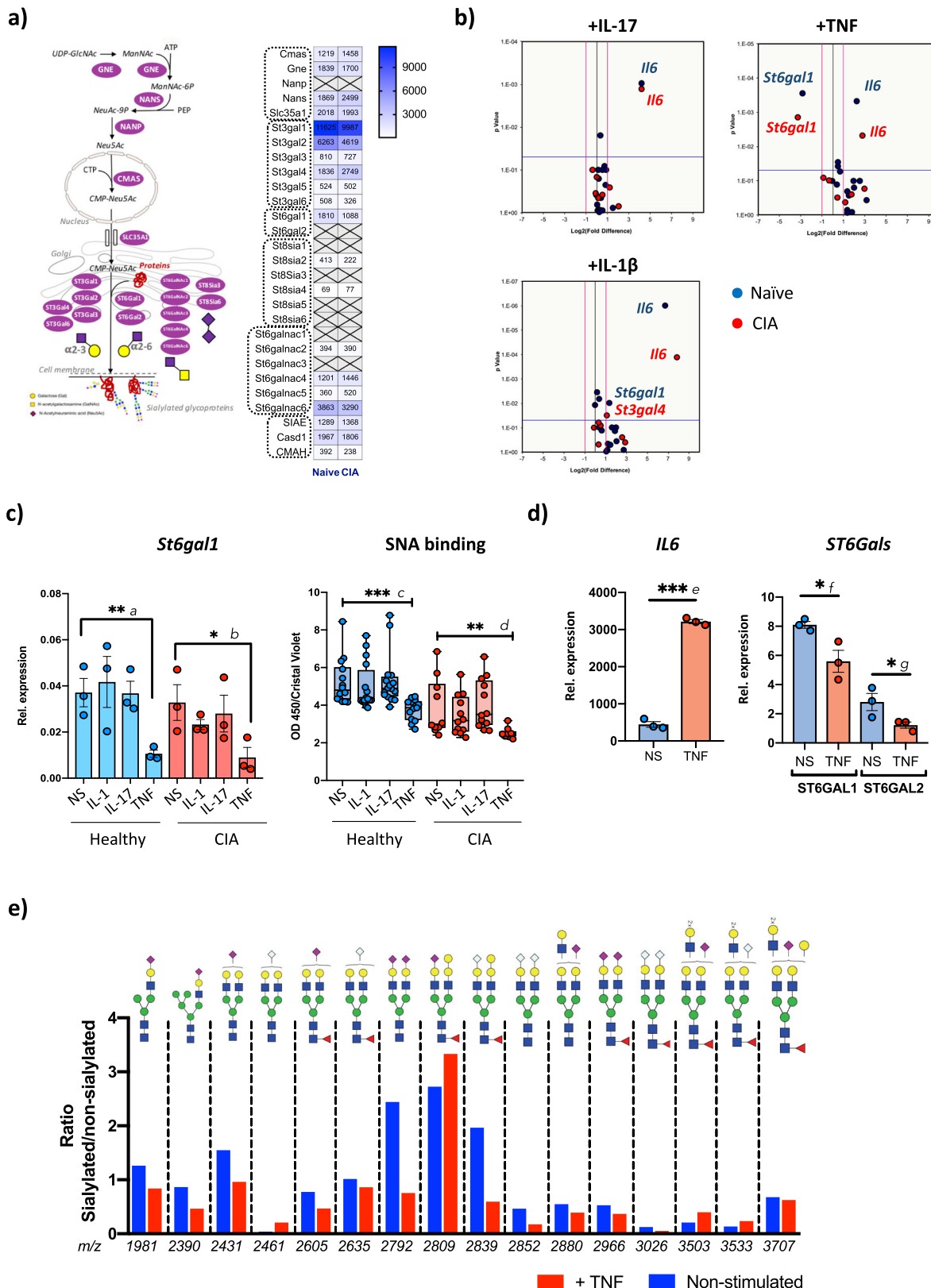

or RA patients in sustained remission under c-DMARDs+TNF-inhibitor combination therapy (Fig. 8). These samples were obtained from US-guided minimally invasive synovial tissue biopsies, which limited the amount of biological material and impacted the detection of low abundant species in the MS-based glycomic studies. Therefore, these cell lines are derived from all

the synovial fibroblasts populations in the tissue. We could still detect the most abundant structures, including high-mannose and sialylated, and non-sialylated bi-antennary N-glycans. Results showed that RA patients in sustained remission had a higher ratio of sialylated/non-sialylated N-glycans, differences that were not seen when other types of structures were compared, such as

**Fig. 5 TNF inhibits St6gal1 expression to down-regulate α2-6-sialylation in synovial fibroblasts. a** Illustration shows pathways involved in sialic acid and sialylated glycans biosynthesis. Relative expression for each gene was extracted from bulk RNA-Seq data comparing naive and arthritic SFs directly isolated from mouse synovium. **b** Synovial fibroblasts expanded from healthy or arthritic mouse joints were stimulated in vitro for 6 h with recombinant IL-1β, TNF, or IL-17A [10 ng/ml]. mRNA expression for genes involved in sialylation [as shown in **a**] was evaluated by RT-qPCR. *Il6* was included as a positive control to confirm cell activation. Volcano plots show log2 fold difference in stimulated cells (*x* axis) and *p*-value (*y* axis). Each dot represents the mean of three independent experiments analyzed in triplicate, with naive cells in blue and arthritic cells in red. The pink line indicates a twofold-change in gene expression threshold and blue lines indicate $p = 0.05$ threshold in the *t*-test to evaluate statistical significance. **c** TNF, but not IL-1β or IL-17A reduces α2-6-sialylation of synovial fibroblasts both at RNA (*St6gal1* expression) and protein (SNA binding) levels. Expression of *St6gal1* (left panel) was evaluated by RT-qPCR. Results show the mean of three independent experiments analyzed in triplicate, error bars represent SEM. Statistical significance was determined using two-tail unpaired *t*-tests where *$p < 0.05$ and **$p < 0.01$. Actual *p*-values: *a* 0.0069, *b* 0.0277. Expression of α2-6 sialylated glycoconjugates was determined by SNA binding (right panel) as in Fig. 4a. Results are merged from 3 (naive) and 2 (CIA) independent experiments, $n = 10$ (DMEM naive), 16 (IL-1 naive and IL-17 naive), 13 (TNF naive), 10 (DMEM CIA), 12 (IL-1 naive and IL-17 naive), and 9 (TNF CIA). Statistical significance was determined using two-tail unpaired *t*-tests, where **$p < 0.01$ and ***$p < 0.001$. Actual *p*-values: *c* 0.0124, *d* 0.0091. **d** SFs were expanded from OA human synovium. Cells were stimulated with recombinant human TNF (10 ng/ml) for 6 h. *IL6*, *ST6GAL1*, and *ST6GAL2* expression were determined by qPCR. Results show relative expression to HPRT, showing mean ± SEM. Statistics: one-tail unpaired *t*-test, $n = 3$ biological replicates from cells pooled from three donors, ***$p < 0.001$, *$p < 0.05$. Actual *p*-values: *e* <0.0001, *f* 0.0170, *g* 0.0321. **e** Relative expression of individual sialylated glycan structures was evaluated by MALDI-TOF MS analysis as in Fig. 3. Ratios of sialylated vs non-sialylated twin structures were evaluated for non-stimulated cells (blue) and TNF stimulated (red) cells (48 h, 10 ng/ml). Results are from one single experiment using pooled cells from three animals.

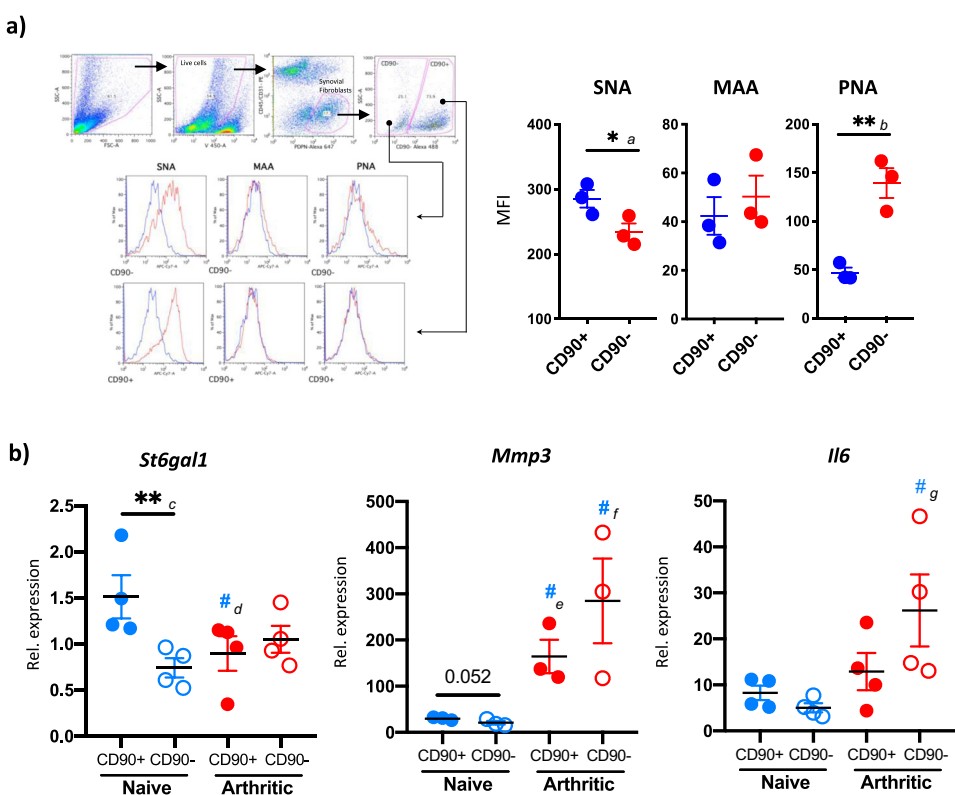

**Fig. 6 Reduced α2-6-sialylation is restricted to CD90⁺ synovial fibroblasts. a** Synovial fibroblasts from healthy mice were isolated from mouse joints and identified by flow cytometry as in Fig. 1 (Zombie Violet⁻, CD45⁻, CD31⁻, Podoplanin⁺), separated into CD90⁺ and CD90⁻ populations and stained with the biotinylated lectins SNA, MAAII, and PNA, and streptavidin-PE-Cy7. Graphs show the mean fluorescence intensity for each lectin. Data are presented as mean ± SEM, dots represent cells from one individual mouse ($n = 3$). One-tail unpaired *t*-test was used for statistics, **$p < 0.01$, #$p < 0.05$. Actual *p*-values: *a* 0.0264, *b* 0.023. **b** Synovial fibroblasts from healthy and arthritic mice ($n = 4$) were isolated from mouse joints as in **a** and RNA was purified using the RNeasy Mini Kit (Qiagen) according to manufacturer's instructions. Expression of *Il6, Mmp3*, and *St6gal1* mRNA was quantified by RT-qPCR. Data are presented as mean ± SEM. One-tail unpaired *t*-test was used for statistics, **$p < 0.01$, #$p < 0.05$ compared to the same population in control healthy mice. Actual *p*-values: *c* 0.0118, *d* 0.0428, *e* 0.0103, *f* 0.0225, *g* 0.0183.

fucosylated versus non-fucosylated or different mannose-containing glycans. This suggests that high sialic acid content is a distinctive feature of SF-glycome in RA patients in sustained remission (Fig. 8a). In line with this, SNA binding revealed a higher expression of α2-6 sialic acid in the synovium of RA patients in remission after c-DMARDs + TNF-inhibitor treatment (Fig. 8b) compared with OA and naive (not exposed to biologic DMARDs) RA synovium, supporting our conclusion of sialylation being an anti-inflammatory factor not only in the mouse model but also in human RA. However, further studies are necessary to determine the effect of SF sialylation in human RA. Given the disease heterogeneity, different disease stages and patient groups must be considered to extract clinically relevant conclusions.

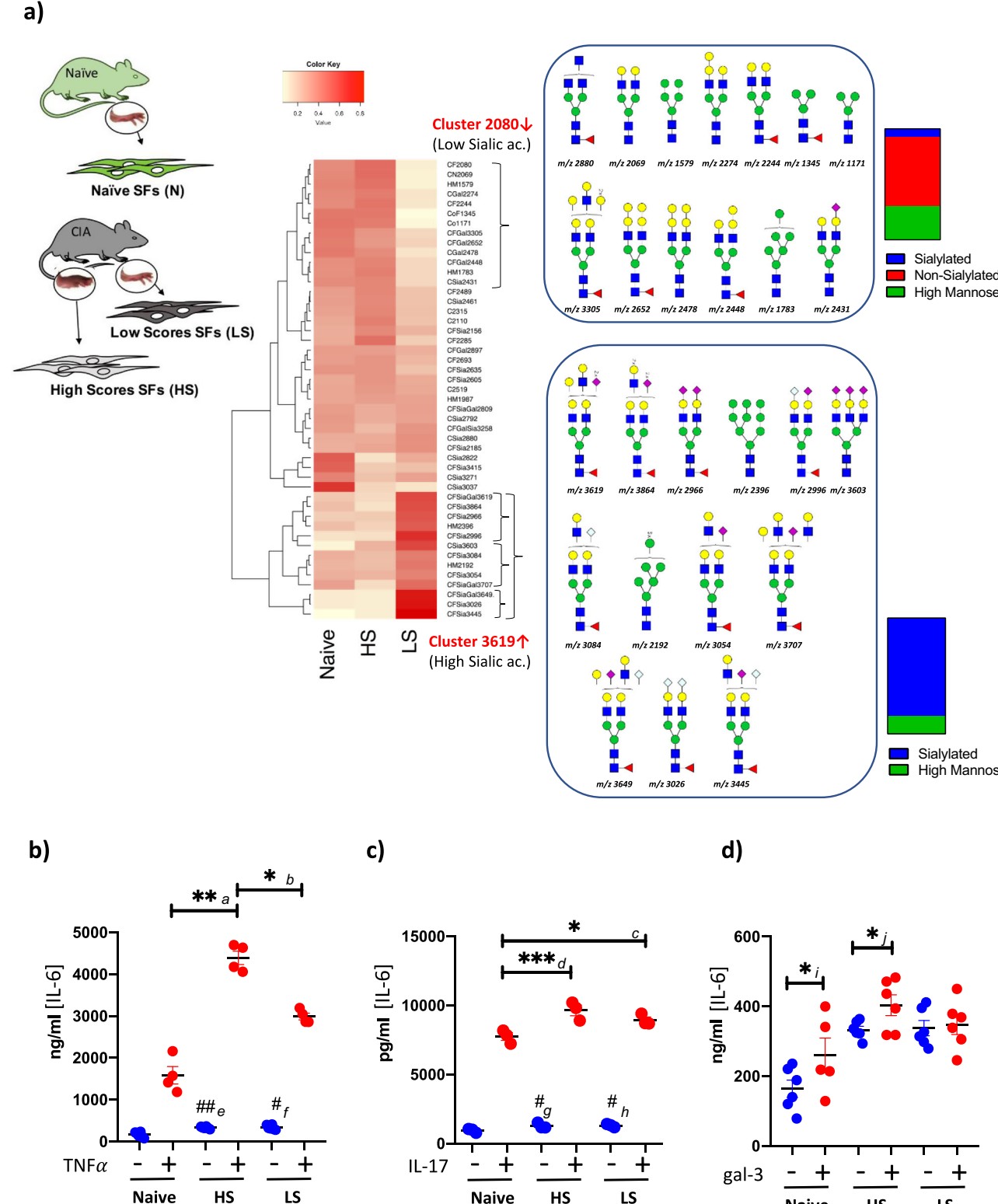

**Exogenous hydrolysis of sialic acid triggers inflammation in SFs**. Prior to clinical studies, it was important to establish whether desialylation plays a leading role in SF activation, or on the contrary, it is a more indirect consequence of ongoing inflammation. If α2-6-sialylation was a molecular switch for controlling inflammatory/resting stages, its removal would trigger pathogenic pathways in naive cells. We conducted experiments to modulate cell sialylation to test this hypothesis. α2-6-sialylation was blocked

by *St6gal1* mRNA silencing, validated by qPCR and SNA binding (Fig. 9a). We observed that siRNA treatment also upregulated *Il6* mRNA (Fig. 9a). The effect of *St6gal1* silencing on cytokine secretion was confirmed by ELISA, as cells produced significantly higher amounts of IL-6 and Ccl2 (Fig. 9b). Increased MMP3 release was also detected, albeit this did not achieve statistical significance (Fig. 9b). Additionally, we used recombinant *Clostridium perfringens* sialidase (CP) to remove surface sialic acid

**Fig. 7 Synovial fibroblasts isolated from inflamed and non-inflamed joints have distinct sialylation signatures.** Synovial fibroblasts were isolated from healthy mice (naive) and from CIA arthritic mice, where joints were separated into non-inflamed (low scores, LS) and very inflamed (high scores, HS) prior to cell isolation and expansion. **a** N-glycans were isolated and permethylated for to MALDI-TOF MS analysis and relative glycan expression was quantified as in Fig. 3. Two clusters were identified as distinctive of severe inflammation, for which glycan structures are shown. Percentage of sialylated and non-sialylated glycans are represented. Cell lines from **a** were stimulated with TNF (**b**) and IL-17 (**c**) [10 ng/ml, 12 h]. Supernatants were collected and IL-6 levels were quantified by ELISA. Each dot represents an independent experiment analyzed in triplicate. Data are presented as mean ± SEM, $n = 3$ (IL-17), and $n = 4$ (TNF) independent experiments. Statistical significance was evaluated by ordinary one-way ANOVA and Tukey's test for multiple comparations. ***$p < 0.001$, **$p < 0.01$, *$p < 0.05$. Actual $p$-values: $a$ <0.0001, $b$ <0.0001, $c$ 0.0265, $d$ 0.0008. #$p < 0.05$ compared with naive non-treated, actual $p$-values: $e$ 0.003, $f$ 0.00028, $g$ 0.0393, $h$ 0.0158. **d** Cell lines were stimulated with galectin-3 [1 ng/ml, 12 h] and IL-6 was measured in culture supernatants by ELISA. Data are presented as mean ± SEM, $n = 6$ independent experiments. One-tail unpaired $t$-test was used for statistics, *$p < 0.05$. Actual $p$-values: $i$ 0.0476, $j$ 0.0243.

from naive SFs. CP hydrolyzed sialic acid in only 30 min, confirmed by the reduced ability of treated cells to bind SNA and MAA, but not AAL (fucose specific) (Fig. 9c). CP reduced SNA binding to 52.6% (±9.1, $n = 6$ experiments) and MAA binding to 64.9% (±23, $n = 6$ experiments) and it had an immediate effect on SF phenotype. CP-treated SFs significantly upregulated *Il6* mRNA (4 fold) and *Ccl2* mRNA (8-fold) after only 3 h, with no difference in *Mmp3* mRNA expression (Fig. 9d). These results corroborate the siRNA experiments (Fig. 9a, b), indicating that a drop of sialic acid in SFs is sufficient to induce *Il6* and *Ccl2* in otherwise resting cells.

## Discussion

Glycans are involved in fundamental biological processes associated with SF-mediated pathophysiology, such as cell adhesion and migration, cell signaling, and communication or immune modulation. There are examples of proteins recognizing glycans to initiate SF-dependent inflammatory responses, such as galectin-3, a positive regulator of TLR-induced responses in human SFs[34]. Although galectin-3 stimulates IL-6, CXCL8 and MMP3 production in both dermal and synovial fibroblasts, it promotes a significantly higher secretion of TNF, CCL2, CCL3, and CCL5 in SFs[19], reflecting distinct stromal glycosylation and immunity of the synovial space. However, little attention has been given to the SFs glycome, the code read by galectins, and all carbohydrate-binding proteins. Yang et al. in 2004 published one of the first studies connecting SF glycosylation and cytokine activity, showing specific glycan remodeling associated with TNF and TGFβ stimulation[35]. However, the structural information generated was very limited due to the nature of the lectin-based assays employed. More recent studies have focused on specific glycosyltransferases expressed in SFs, like fucosyltransferase 1 (Fut1) and galactosyltransferase-I (β1,4-GalT-I). Fut1 is involved in terminal α1–2 fucosylation and is upregulated in RA synovial tissue[36] regulating leukocyte-SF adhesion, whilst β1,4-GalT-I is induced by TNF[37] to promote binding to the glycosylated extracellular matrix. Aberrant glycosylation of fibronectin has also been reported in RA[38]. Nonetheless, the exact composition of the SF-glycome was unknown, limiting the progression of functional glycomics studies in joint inflammation. Therefore, we aimed to conduct a broader systematic study to bridge the gap between glycomics and other omics approaches in the field of SF-dependent immunology. Our results provide an extensive description of the SF glycome in models of health and disease that will help to understand the interactions of SFs with other immune cells and matrix components in the joint tissue in human disease.

The success of glycosciences in the field of cancer is an exemplar model that integrates glycomics into stromal immunology. Distinct modifications in tumors have been characterized including truncated N- and O-glycans, increased N-glycan branching, and changes in sialylation. Ultimately, these changes affect fundamental cell processes like cell adhesion, signaling,

tumor progression, and metastasis[23], making glycans attractive targets for therapeutic intervention. Recent technological advances and the prospective use of glycan-based biomarkers for patient stratification have led it to be estimated that the global glycobiology market size will grow 14.7% in the coming years, with an estimated market value of USD 822.5 million in 2018. In RA, SFs undergo important epigenetic changes to adopt a tumor-like invasive phenotype that is also likely to be determined by glycosylation. This may include hyperproliferation, secretion of MMPs responsible for tissue damage and migration, and perpetuation of local pro-inflammatory responses. Thus, perhaps new therapeutic opportunities in RA can be opened by mirroring recent advances in cancer, especially for patients that are refractory to current immunosuppressive drugs. A better understanding of SF-glycobiology will aid the development of such glycan-based therapeutics.

We combined transcriptomic and MS-based glycomic analysis to define changes in SF-glycome with unprecedented detail in synovial inflammation. N-glycome from healthy SFs comprises LacNAc-containing structures with high levels of core fucosylation and terminal sialylation. However, inflammation renders the sub-lining CD90+ SF population hyposialylated, specifically in α2,6-linkage. Differences between the lining and sub-lining sialylation could be explained by distinct expression and regulation of TNF receptors and associated signaling molecules. Our data include all CD90+ populations, but further differences can be expected in sub-lining fibroblasts subsets. Hyposialylated SF-glycome would increase the accessibility of LacNAc repeats, allowing galectin-3 binding to SF surface glycoconjugates. In fact, sialylation can act as a "switch off" for galectin-3 function. For example, ST6Gal1 up-regulates α2-6-sialylation to block binding of galectin-3 to β1 integrin, inhibiting cell apoptosis[39]. Sialylation also inhibits galectin-3 binding to squamous epithelia and tumor cells[40–42]. Thus, because of a desialylated glycome, galectin-3 could induce secretion of inflammatory mediators such as IL-6 or Ccl2 in SFs, increasing local concentrations of TNF that would further reduce ST6Gal-1 and α2-6-sialylation, establishing an autocrine pro-inflammatory loop that could explain the perpetuation of disease in a similar fashion shown by members of the IL-6 family[43]. By contrast, healthy synovium shows higher levels of α2-6-Sialylation in SFs, perhaps masking LacNAc repeats and preventing inflammatory actions of galectins. Likewise, the spontaneous upregulation of *Il6* and *Ccl2* when sialic acid is directly removed from the SF surface could be a consequence of galectin signaling in an autocrine fashion. Additionally, other glycan modifications may play a role, such as a glycan branching. Based on the collective evidence provided here, we propose that sialylation might be a homeostatic mechanism that is lost during RA progression because of TNF overexposure.

To understand the physiological role of sialic acid in SFs, the role of additional carbohydrate-binding proteins (CBPs) will require consideration. CBPs are expressed either in SFs, or in

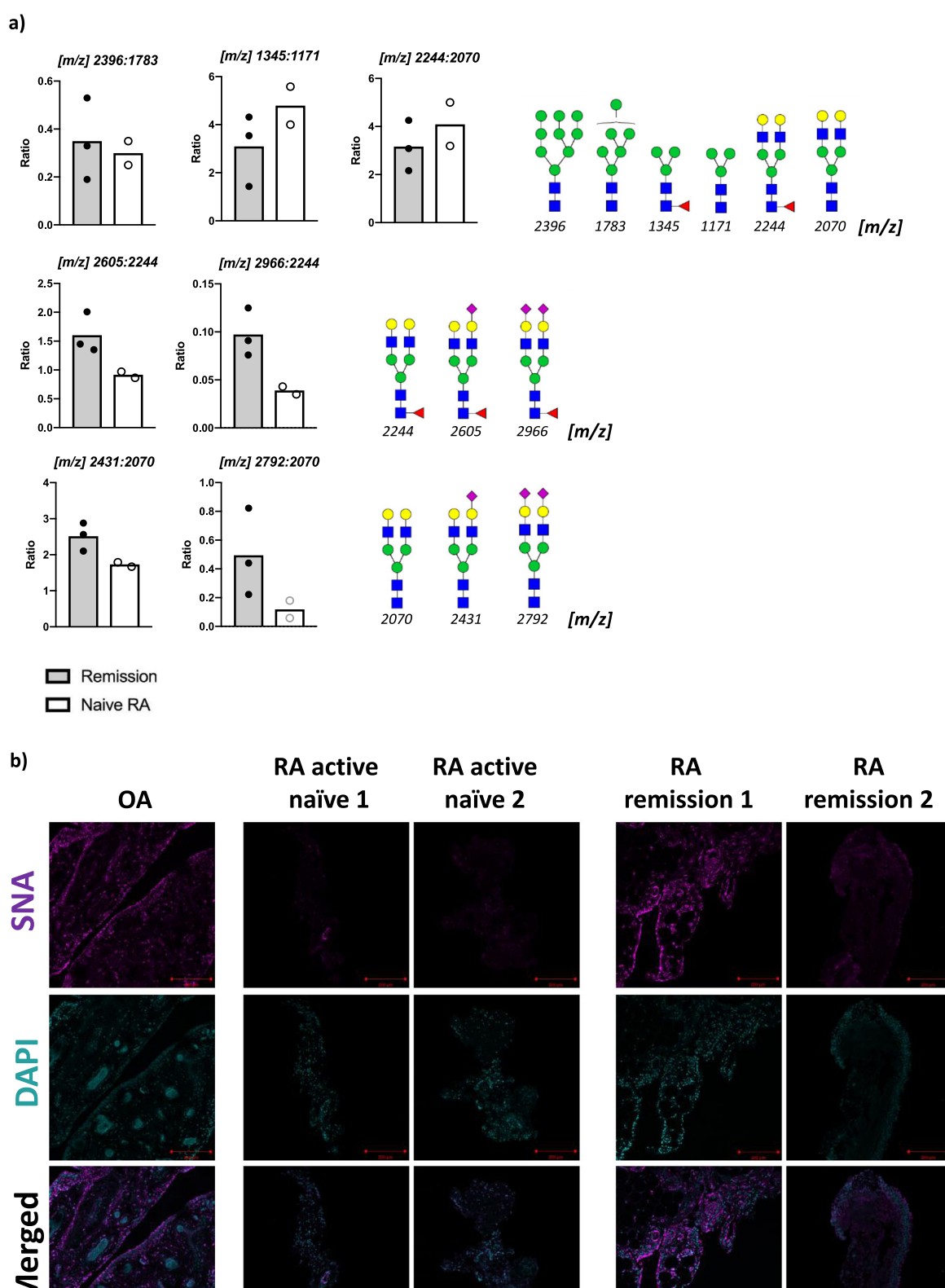

**Fig. 8 Levels of sialylation correlate with disease stages and pathotypes of human RA. a** Synovial fibroblasts were expanded from human biopsies from $n = 2$ naïve RA patients (not exposed to any disease-modifying anti-rheumatic drug) and $n = 3$ RA patients in sustained clinical and imaging remission achieved with c-DMARDs+TNF-inhibitor combination therapy. N-glycans were isolated and subjected to MALDI-TOF MS analysis. Relative intensity was used to calculate the ratio of expression of the indicated paired glycans. Glycan structures are designated by their *m/z* value. **b** Synovium from OA, and naïve and remission RA patients were stained with biotinylated SNA and Alexa-647-conjugated streptavidin (magenta). DAPI (cyan) was used to stain cell nuclei. Images were acquired with a confocal microscope LSM 880. Results are from one single experiment, OA $n = 1$, active RA = 2, remission RA $n = 2$. Each image represents one individual patient. Scale bars: 200 μm.

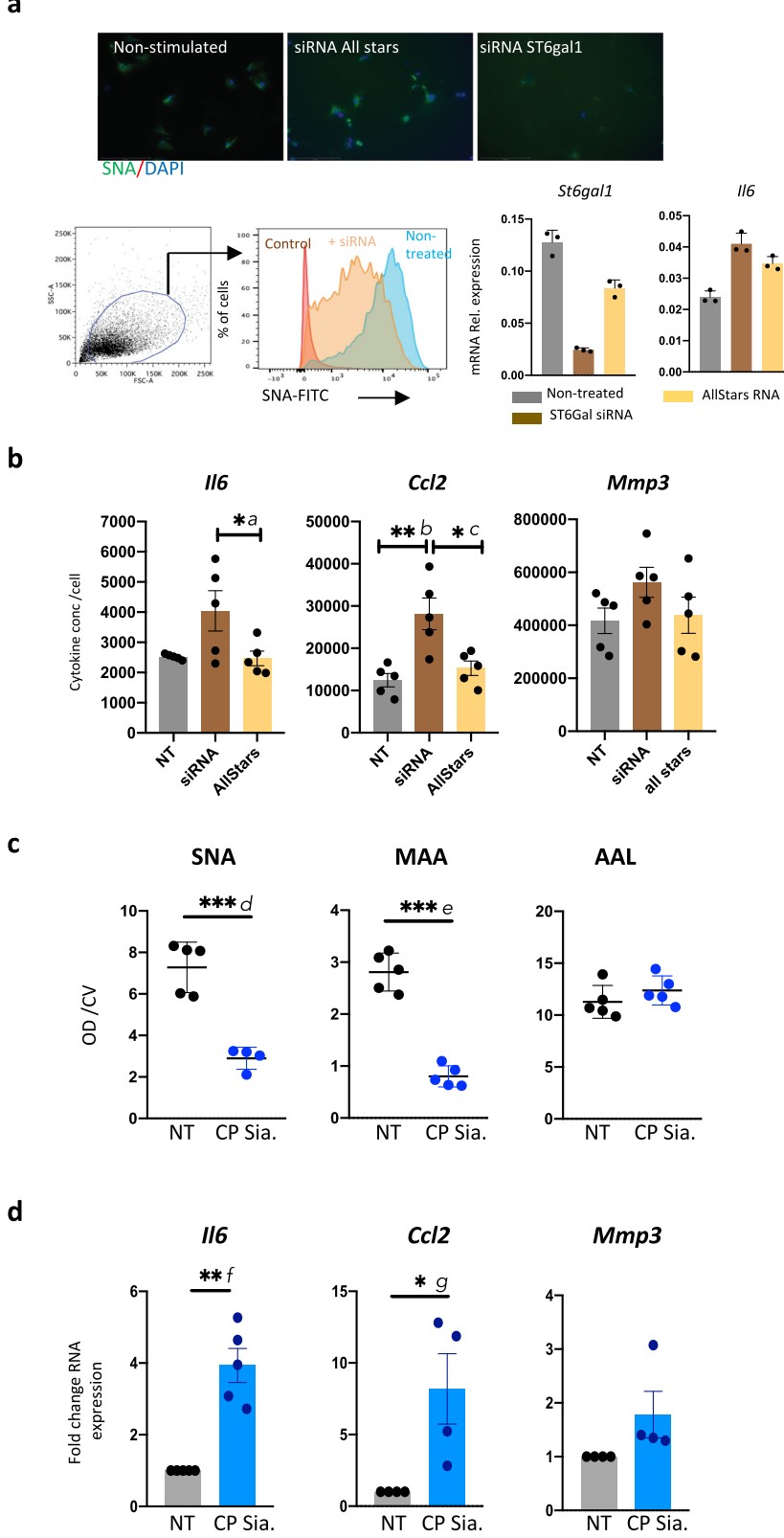

other immune cells, promoting trans or cis interactions. Siglecs (sialic acid-binding immunoglobulin-type lectins) are a family of immune regulatory proteins primarily found on hematopoietic cells. Consistent with our proposed homeostatic role for sialic acid in the synovium, Siglec-9 protected mice against experimental arthritis, although authors reported that it had no effect on SFs[44]. Instead, Siglec-9 inhibited NF-kB activation in human

RA macrophages. However, effects on SFs cannot be ruled out, as following cytokine stimulation during disease, RASFs may already have a reduced sialylation profile which would make them unresponsive to Siglec-mediated actions. It is, therefore, possible that siglecs mediate regulatory effects in synovial self-tolerance under healthy conditions, or at very early disease stages. The mechanism could involve Siglec-sialic acid trans interactions

**Fig. 9 Synovial fibroblasts desialylation triggers inflammatory cytokine production. a** Murine SFs isolated from healthy mice were treated with either siRNA for *St6gal1* or Allstars RNA control for 3 days. The reduction of α2-6-sialic acid was evaluated by SNA binding by immunofluorescence and flow cytometry. Expression of *St6gal1* mRNA and *Il6* mRNA was determined by qPCR. **b** SFs were treated with *St6gal1* siRNA or Allstars RNA for 3 days. Cell culture medium was then replaced and cells were incubated for 12 h at 37 °C. Cytokine concentration was determined by ELISA and data were normalized with the number of cells in each well, calculated by crystal violet staining. Each dot represents one experiment analyzed in technical triplicates. Statistical significance was evaluated by ordinary one-way ANOVA and Tukey's test for multiple comparations, *$p < 0.05$, **$p < 0.01$. Actual $p$-values: *a* 0.0474, *b* 0.025, *c* 0.100. **c** murine SFs were incubated with 100 mU/ml of recombinant *Clostridium perfringens* (CP) sialidase for 30 min at 37 °C. Cells were then washed and fixed. Relative lectin binding (SNA, MAA, and AAL) was then calculated by ELLA assay. Data are from one representative experiment performed in five biological replicates. The experiment was repeated four times with similar results. ***$p < 0.001$ analyzed by one-tail *t*-test. Actual $p$-values: *d* 0.0003, *e* <0.0001. **d** Synovial fibroblasts were treated with 100 mU/ml of recombinant *Clostridium perfringens* (CP) sialidase as before and incubated for 3 h at 37 °C. RNA was then isolated to quantify relative expression levels of *Il6, Ccl2,* and *Mmp3* by qPCR. Data represent fold change expression compared to non-treated controls. Each dot represents one independent experiment analyzed in triplicate, data are presented as mean ± SEM for $n = 5$ experiments for *Il6* and *Ccl2* and $n = 4$ experiments for *Mmp3*. A two-tail paired *t*-test using relative expression was used to calculate statistical significance. *$p < 0.05$, **$p < 0.01$. Actual $p$-values: *f* 0.0034, *g* 0.0265.

between SF and immune cells. In line with this, B cells lacking Siglec-2 (CD22) and Siglec-G develop spontaneous autoimmunity[45], and Siglec-G$^{−/−}$ lupus-prone MRL/lpr mice exhibit increased severity and early onset of arthritis[46]. Besides, most siglecs (2,3,5,6-11) show inhibitory effects on TLR-dependent activation and mediate immunosuppression in the tumor microenvironment because of local hypersialylation[47]. By contrast, Siglec-1 (sialoadhesin) shows pro-inflammatory actions, it is upregulated in activated macrophages in RA[48] and suppresses Tregs[49]. Interestingly, the anti-inflammatory Siglec-2 has a predilection for α2-6 sialic acid, but the pro-inflammatory Siglec-1 binds preferentially to α2-3[50], in turn supporting our conclusion that pro-inflammatory SFs diminish α2-6-sialylation but not α2-3. Thus, homeostatic α2-6 >>> α2-3-sialylation on SFs could induce Siglec-2-mediated signals to prevent B cell activation, whereas inflammatory α2-6 < α2-3-sialylation would support SFs interactions with pathogenic macrophages via Siglec-1, cells that also secrete high levels of pro-inflammatory galectin-3 and TNF, perpetuating disease. Although this is an oversimplification of a complex scenario, it provides a good example of how the axis cytokines-sialylation-siglecs/galectins could regulate interactions between immune cells and SFs. Moreover, other sialic acid-binding proteins could be relevant, like selectins. Selectins are expressed in lymphoid, myeloid, and endothelial cells, and mice deficient in Selectin-P and selectin-E show an enhanced arthritis progression[51,52]. Conversely, ficolins, innate immune receptors recognizing sialic acid too, show opposite effects, as ficolin B deficient mice are protected against arthritis[53,54]. These glycan-dependent networks could explain the SF-mediated pathology and their interactions with immune cells in the arthritic synovium. Further work should be directed to determine the expression of sialic acid-binding proteins in different cell types in the synovium to complement the findings presented in this work.

Pathogenic networks will be further defined by the inclusion of additional immune cells in the equation, but also by the SFs anatomical and functional heterogeneity that has been recently revealed[10,30]. Sub-lining CD90$^+$ SFs are the main contributors to inflammatory responses in RA, observation validated in animal models[9]. Our results in the mouse model indicate that only sub-lining CD90$^+$ SFs reduce *St6gal1* expression, compared to the lining CD90$^−$ SFs suggesting that sialic acid would have distinct roles on individual SFs subsets. These sialoside-dependent interactions could explain the observed disease tissue heterogeneity described in RA patients, including lymphoid, myeloid, and fibrotic phenotypes[55,56]. Because synovial phenotypes observed in RA have been associated with distinct pathways (myeloid-IL-1β/TNF, lymphoid-IL-17[55]), it would be relevant to study ST6Gal1 expression in RA phenotypes. Interestingly, TNF induces a similar shift of α-2,6/α-2,3-sialylation in chondrocytes

downregulating ST6Gal1[57], in agreement with the pro-inflammatory role of α2,3-sialylation in synovial joints.

In conclusion, in the present report, we revealed that the reduction of α2,6-sialylation constitutes a signature of the inflammatory status of synovial fibroblasts. This altered phenotype seems to be induced by TNF, in contrast with IL-1 or IL-17 that had no effect. This might open a possibility to segregate disease phenotypes by their glycosylation profile. Future studies clarifying the functional consequences of shifting sialylation patterns in SFs are now required, especially in the clinical context, as well as expressions of sialic acid receptors in other cells, like B cells or monocytes. The vast, and yet unexploited amount of information contained in the SF glycome could offer novel therapeutic targets, and further work is anticipated before this goal can be achieved. Nevertheless, this study sets the foundations for future clinic intervention of glycan-dependent pathological networks in human RA, and other autoimmune diseases where stromal fibroblasts control local inflammation.

## Methods

**Patient recruitment and isolation of human fibroblasts lines**. Tissue samples were collected from patients who fulfilled the 2010 EULAR/American College of Rheumatology classification criteria for RA[58]. Tissue samples were also acquired from patients with radiographically confirmed OA. Synovium samples and overlying skin were obtained from the knee joint of each patient at the time of joint replacement surgery. This study was reviewed and approved by the South Birmingham Local Ethics Committee (LREC 5735), all patients gave written informed consent. Tissue samples were minced into 1 mm$^3$ section under sterile conditions, washed, and then resuspended in 10 ml 5 mM EDTA in phosphate-buffered saline (PBS) and incubated for 2 h at 4 °C with vigorous shaking. The resulting cell–tissue mixture was washed 3 times in RPMI medium and then cultured in complete RPMI 10% FCS until adherent fibroblast colonies became confluent. Non-adherent cells and tissue fragments were discarded once adherent cells had appeared. Fibroblasts were expanded until reaching confluence and then reseeded into tissue culture flasks of twice the surface area after trypsin treatment. All experiments using expanded fibroblast lines used cells between passages 3 and 4. For active naive RA patients and patients in sustained clinical and ultrasound remission[59], biopsies were collected at the Division of Rheumatology of the Fondazione Policlinico Universitario A. Gemelli IRCCS – Università Cattolica del Sacro Cuore (namely SYNGem cohort) through ultrasound guidance using a standard operating procedure with a sampling of at least 6–8 different tissue pieces[60,61]. Patient demographics are included in Supplementary Table 2.

**Immunofluorescence analysis**. Overall, 10,000 cells were grown in chamber slides, wash three times with PBS, and fixed with 4% PFA for 20 min at room temperature. Cells were permeabilized with PBS 0.5% Triton and rinsed with PBS-Tween 20. Endogenous biotin was blocked with the Streptavidin/Biotin blocking kit according to the manufacturer's instructions (Vector Laboratories, SP-2002). Carbo-Free Blocking Solution (Vector laboratories, SP-5040) was used as a blocking solution. Samples were incubated overnight with biotinylated lectins at 2 μg/ml in PBS purchased from Vector laboratories [Concanavalin A (ConA), B-1005; Ricinus Communis Agglutinin I (RCA), B-1085; Peanut Agglutinin (PNA), B-1075; Jacaline, B-1155; Wheat Germ Agglutinin (WGA), B-1025; Sambucus Nigra Lectin (SNA), B-1305; Lycopersicon Esculentum Lectin (LEL), B-1175; Erythrina Cristagalli Lectin (ECA), B-1145; Aleuria Aurantia Lectin (AAL), B-1395;

Ulex Europaeus Agglutinin I (UEA), B-1065]. Lectins were finally detected with FITC-conjugated streptavidin (Biolegend, 405201). Mouse joint sections (7 μm) and human synovium sections were deparaffinized in xylene and dehydrated in ethanol, and antigen was retrieved by incubation at 60 °C for 40 min in sodium citrate buffer (10 mM Sodium Citrate, 0.05% Tween 20, pH 6.0). Blocking was performed as explained before. Samples were stained with biotinylated Phaseolus Vulgaris Leucoagglutinin (PHA-L) from Vector laboratories, catalog number B-1115, and Alexa-555-conjugated streptavidin (ThermoFisher, S21381). Vimentin was detected with anti-vimentin goat IgG (Sigma, V4630, diluted 1:400) and Alexa-647-conjugated anti-goat IgG (ThermoFisher, A21245, diluted 1:1000). All samples were counterstained with DAPI. Images were obtained using an LSM 510 META confocal laser coupled to an Axiovert 200 microscope (Zeiss) and analyzed with Zeiss LSM Image Browser software 4.0.

**Flow cytometry**. Cultured cells were detached from the plates using Accutase cell detachment solution (Stemcell Technologies) according to the manufacturer's instructions. For lectin staining (listed before), cells were blocked with Carbo-Free Blocking Solution (Vector laboratories, SP-5040) and stained with biotinylated lectins and FITC-conjugated streptavidin in PBS for 20 min at 4 °C. Data were acquired using a BD LSRII flow cytometer and analyzed with FlowJo software 8.7.3.

**Mice and CIA model**. CIA was induced in 8–10-week-old male DBA/1 mice (Envigo) on day 0 by intradermal immunization with bovine type II collagen (MD Biosciences) in Freund's complete adjuvant (CFA). On day 21, mice received 100 mg of collagen in PBS intraperitoneally. Disease scores were measured every 24 h on a scale from 0 to 4 for each paw. Animals reaching an overall score of 10 or more were immediately euthanized. Animals were maintained in the University of Glasgow Biological Services Units in accordance with the Home Office UK Licenses P8C60C865, I675F0C46, and ID5D5F18C, and the Ethics Review Board of the University of Glasgow. Mice were maintained under 12 h light/dark cycles and standard temperature (20–25 °C) and humidity (40–50%).

**Isolation of synovial fibroblasts from mice and explant cell culture**. For mouse synovial tissue digestion, skin and soft tissues were removed from mouse limbs, and bones with intact joints were dissected and transferred into DMEM (+5% FCS) containing 1% L-Glutamine, 1% Penicillin Streptavidin, nystatin, and 1 mg/ml type IV collagenase (Sigma) and 5 mg/ml DNase I. Samples were incubated in the shaking oven at 37 °C for 1:20 h, when EDTA was added, final concentration 0.5 mM for 5 min at 37 °C. Samples were vortexed to release cells. Cells were centrifuged and washed with DMEM 10% FCS twice. For cell expansion, cells were plated and expanded until adherent fibroblast colonies became confluent. Non-adherent cells and tissue fragments were discarded once adherent cells had appeared, usually after 48 h. Fibroblasts were expanded until reaching confluence and then reseeded into tissue culture flasks after trypsin treatment. Expanded synovial fibroblasts were used at passage 3 or 4 only, when culture purity was assessed by flow cytometry and expression of CD106 (Biolegend, catalog number 105722), CD54 (Biolegend, catalog number 116107), and CD90 (Biolegend, catalog number 105316). Expression of CD11b was also evaluated (Invitrogen, catalog number 11-0112-85), where cells were <1% positive. All antibodies were used at 1:100 dilution.

**Mouse synovial fibroblast sorting**. Single-cell suspensions from mouse synovium were obtained as described before. Cells were stained at 4 °C with Zombie Violet staining (BioLegend, catalog number 423113) to exclude dead cells. Antibodies used were anti-CD45 (Biolegend, catalog number 103106), anti-CD90 (Biolegend, catalog number 103106), anti-podoplanin (Biolegend, catalog number 105316), anti-CD31 (Invitrogen, catalog number 12-0311-81). Cell sorting was performed immediately after staining using a FACS Aria Ilu machine. For sorted populations, purity was determined by reanalysis for the target population based on cell surface markers immediately post sorting. Purity was >99% for the synovial fibroblasts target population (CD31⁻, CD45⁻, podoplanin⁺). All antibodies were used at 1:100 dilution.

**RNA-sequencing and data analysis**. Total RNA from sorted synovial fibroblasts was isolated immediately post sorting using RNeasy Micro kit (Qiagen, Germany). RNA integrity was checked with the Agilent 2100 Bioanalyzer System. All purified RNA had a RIN value >9. Libraries were prepared using the TruSeq mRNA stranded library preparation method. Samples were sequenced 2 × 75 bp to an average of more than 30 million reads. The data discussed in this publication have been deposited in NCBI's Gene Expression Omnibus[62] and are accessible through GEO Series accession number GSE162306. All RNA-seq reads were then aligned to the mouse reference genome (GRCM38) using Hisat2 version 2.1.0. Featurecounts version 1.4.6 was used to quantify reads counts. Data quality control, non-expressed gene filtering, median ratio normalization (MRN) implemented in DESeq2 package, and identification of differentially expressed (DE) genes was done using the R Bioconductor project DEbrowser[63]. Genes that passed a threshold of $p_{adj} < 0.01$ and log2foldChange > 2 in DE analysis were considered for further analysis. Gene Ontology (GO) enrichment, KEGG pathway enrichment, and

UniProt Keywords enrichment were performed in String version 11.0 (https://string-db.org) based on statistically significant DE genes.

**Mass spectrometric analysis of glycans**. Synovial fibroblasts cells were scraped off tissue culture plates and suspended in iced-cold ultrapure water before homogenization and sonication were performed. Cells protein extract was precipitated in a methanol/chloroform extraction, Cell extracts were reduced and carboxymethylated, using Dithiothreitol and Iodoacetic acid, and then treated with trypsin. The treated samples were purified using Oasis HLB cartridges (Waters) prior to the release of N-glycans by PNGase F (recombinant from Escherichia coli, Roche) digestion and O-glycans were by reductive elimination. Released glycans were permethylated and then purified using a Sep-Pak C18 cartridge (Waters) prior to MS analysis. The resulting enzyme-treated samples were lyophilized and permethylated prior to MS analysis. Purified permethylated glycans were dissolved in 10 μl methanol and 1 μl of the sample was mixed with 1 μl of matrix, 20 mg/ml 2,5-dihydroxybenzoic acid (DHB) in 70% (v/v) aqueous methanol and loaded on to a metal target plate. 4800 MALDI-TOF/TOF mass spectrometer (AB SCIEX) was run in the reflectron positive ion mode to acquire data. MS spectra were annotated manually with the assistance of the glycobioinformatics tool GlycoWorkBench (GWB) version 1.1. All N-glycans were assumed to have a Manα1–6(Manα1–3) Manβ1–4GlcNAcβ1–4GlcNAc core structure based on knowledge of the N-glycan biosynthetic pathway in mammalian cells and PNGase F specificity. The composition of the glycans derived from MALDI-TOF MS in positive ion mode was manually interpreted. Relative expression of individual glycan structures was evaluated by calculating areas under the curve of annotated peaks using GWB[64] 1. For comparative studies, MS peak areas were quantified and normalized against total measured intensities. Unsupervised hierarchical clustering with Euclidean distance was performed on the relative glycan expression samples using the heatmap.2 function of the gplots package in R. The symbolic nomenclature for glycan structures used in all the spectra annotations is the same as the one used by the Consortium for Functional Glycomics (CFG) [http://www.functionalglycomics.org/static/consortium/Nomenclature.shtml. Hexoses: circles, N-Acetylhexosamines: squares. Galactose stereochemistry: yellow, Glucose stereochemistry: blue, Mannose stereochemistry: green. Fucose: red. Acidic sugars: diamonds, NeuAc (purple), and NeuGc (light blue)].

**Enzyme-linked lectin assays and ELISA**. Synovial fibroblasts at 10,000 cells/well were grown on 96-well plates. Cells were washed three times with cold PBS. Cells were then fixed with 4% paraformaldehyde for 20 min. Fixed cells were washed three times with PBS containing 0.05% Tween. CarboFree solution (Vector Laboratories) was used to block non-specific interactions, following incubation with 1 μg/ml of biotinylated lectin (Vector Laboratories, Burlingame) for 30 min. Cells were washed with PBS and incubated with HRP conjugated streptavidin for 20 min and washed again with PBS-Tween. A reaction was induced in the cells with a developing solution, consisting of 1 mg/ml p-nitrophenyl phosphate in 0.5 mmol/l. The reaction was allowed to proceed in the dark and plates were read at 405 nm using a microplate spectrophotometer. Interleukin-6 (IL-6) expression was measured by ELISA in SF-supernatants according to the manufacturer's instructions (BD Biosciences, Oxford, UK).

**qRT-PCR**. Cells were lysed in RLT lysis buffer prior to mRNA extraction using RNeasy Plus Mini kit (Qiagen, Germany) according to the manufacturer's instructions. The High Capacity cDNA Reverse Transcriptase kit (Applied Biosystems, Life Technology, UK) was used to generate cDNA for use with StepOne PlusTM real-time PCR system (Applied Biosystems, UK) and predesigned KiCq-Start® qPCR Ready Mix (Sigma-Aldrich) or TaqMan gene expression assays (Applied Biosystems). Predesigned KiCqStartTM primers (Sigma-Aldrich) were purchased to evaluate murine *β-Actin, Cmas, Gne, Nanp, Nans, Slc35a1, St3gal2, St3gal3, St3gal4, St3gal5, St3gal6, St8sia1, St8sia2, St8sia3, St8sia4, St8sia5, St8sia6, St6galnac2, St6galnac4, St6galnac2, St6galnac4, St6galnac5* and *St6galnac6*. Sequences can be found in Supplementary Table 1. TaqMan predesigned probes (ThermoFisher Scientific) were used to evaluate mouse Actb (4352933E), *Il6* (Mm00446190_m1), St6gal1(Mn00486119_m1), St6gal2(Mm01268915_m1), St3gal1(Mm00501493_m1), St6galnac1(Mm01252949_m1), St6galnac3 (Mm01316813_m1) and *Mmp3*(Mm00440295_m1) and human *IL6* (Hs00174131_m1), ST6GAL1 (Hs00949382_m1), ST6GAL2(Hs00383641) and HPRT(4333768T). Data were normalized to the reference gene β-actin in mouse cells and HPRT in human samples to obtain the ΔCT values that were used to calculate the fold change from the ΔΔCT values following normalization to a biological control group.

**Desialylation of synovial fibroblasts in vitro**. For short interfering RNA transfection, St6gal1 siRNA (Qiagen, catalog # SI01434699) or Allstar negative control siRNA (Qiagen, catalog # SI03650318) were diluted in 8% HiPerFect Transfection Reagent (Qiagen, catalog # 301705) in Dulbecco's Modified Eagle Medium (Invitrogen, catalog # 21969035) and incubated 10 min at room temperature to form transfection complex before adding to cells. The final concentration of siRNA for all targets was 10 nM. Cells were then incubated under growth conditions for 24 h before refreshing medium. Follow-up experiments were carried out three days after

the initial transfection. For hydrolytic cleavage of sialic acid, synovial fibroblasts were cultured until 80% confluence. Cells were washed with DMEM and incubated for 30 min at 37 °C with 100 mU/ml of recombinant *Clostridium perfringens* sialidase (Sigma-Aldrich) in PBS:RPMI 1640 (1:1) pH = 6.8. Cells were then washed with 10% FCS DMEM.

**Reporting summary**. Further information on research design is available in the Nature Research Reporting Summary linked to this article.

## Data availability

Transcriptomic data, including raw sequencing data and processed read counts for each sample, have been deposited in NCBI's Gene Expression Omnibus under the accession code GSE162306. Human data shown in Supplementary Fig. 2 were obtained and analyzed from the publicly available data set from https://immunogenomics.io/fibrotime. Raw RNA-Seq data was accessed through GEO accession code GSE129488. All other data are available in the article and Supplementary files or from the corresponding authors upon reasonable request. Source data are provided with this paper.

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

## Acknowledgements
This work was funded by a Career Development award to M.A.P. from Versus Arthritis (21221).

## Author contributions
M.A.P. conceived and oversaw the project, interpreted all the results, and wrote the manuscript with feedback from all authors. M.A.P. and Y.W. performed glycan preparation for MS, tissue culture, ELLA experiments, qPCR, and lectin staining of human synovial fibroblasts. Y.W. performed cell transfections and ELISAs, cell culture, flow cytometry, qPCR, tissue immunohistochemistry, and immunofluorescence. M.A.P., Y.W, and A.K. conducted animal models, FACs-sorted murine synovial fibroblasts, and analyzed RNA-seq data. Y.W. and A.K. isolated RNA for RNA-Seq experiments. K.P.-L. and M.K.-S. helped with data interpretation. S.A., E.G., and B.T. performed synovial biopsies for RA remission samples and active controls and ran synovial tissue processing. A.A. and L.B. performed MALDI sample preparation of derivatized glycans, acquired MS data, and helped with annotation and interpretation of all MS glycomic data. C.D.B., A.F., and K.R. provided the protocol for human synovial samples and culture, helped with data interpretation and experimental design and writing of the manuscript, processed human synovium samples from arthroplasties, and isolated and expanded fibroblasts. A.D. provided a protocol for N-glycan and O-glycan isolation and helped with annotation and interpretation of all MS glycomic data. S.H. helped with annotation and interpretation of all MS glycomic data, contributed to the experimental design of glycomic experiments, and supervised integrative analysis of glycomic and transcriptomic data. All authors approved the manuscript.

## Competing interests
The authors declare no competing interests.
