## [Peer Review File · Nature Communications]

REVIEWER COMMENTS

Reviewer #1 (Remarks to the Author):

In this paper Pineda and colleagues explore sialylation of RA and CIA synovial fibroblasts. TNF is held responsible for alpha-2-6 hyposialylation of CD90+ sublining FLS, and this may be a marker, and perhaps even a cause for aggressive behavior of the FLS. This work is interesting but complex and occasionally paradoxical or puzzling. I have a few specific comments.

1- Why are the changes in sialylation confined to one subset of FLS?

2- To what extent do the glycomic patterns from the FLS signify changes in glycolipids versus glycoproteins?

3- The range of functional studies of the FLS in this paper is rather limited. Prior observations from other groups do not suggest that IL-6 hyper-production and proliferation/migration/cartilage invasion are performed by the same FLS subsets.

4- Are the RA FLS lines developed for this study (from patients with active disease) CD90+ and exclusively representative of the sublining, hyposialylated FLS?

5- PHA also binds strongly to T lymphocytes. In the tissue sections how is lymphocyte binding distinguished from FLS binding?

6- The writing is uneven, Some of the sections are not well stitched together. For example, IL-22 appears out of nowhere and then disappears. Irritating phrases such as "an under-sialylated and highly pro-inflammatory microenvironment that contributes to an amplificatory inflammatory network that perpetuates chronic inflammation" are encountered, much to the displeasure of the reader.

Reviewer #2 (Remarks to the Author):

Wang Y and coauthors report the in-depth investigation into the glycosylation differences between various fibroblasts from patients with rheumatoid arthritis or osteoarthritis, as well as those from an approximate mouse model (collagen-induced arthritis) in which cytokine treatment could be performed. Glycosylation was determined by lectin interaction and mass spectrometry and the authors coupled these to transcript analysis and cytokine intervention, ultimately arriving at the conclusion that alpha-2,6-linked sialylation is decreased in fibroblasts from an arthritic background.

The presented body of work is impressive, as are the suggestions provided thereby. However, I do have several reservations about the interpretation of the lectin and glycomics experiments, as well as about eagerness with which biological conclusion are drawn. My specific comments can be found below.

1 - The manuscript title suggests that the loss of alpha-2,6-linked sialylation causes the development of proinflammatory synovial fibroblasts, but the data in the manuscript suggests only a correlation between the two at best. To claim causation the authors have to include experiments with targeted modification of the glycosylation.

2 - Figure 1 does not generally support the conclusions of the authors as no apparent distinction can be seen in WGA and SNA staining between fibroblast locations. Can the authors explain this discrepancy?

3 - There are several profound differences between human and murine glycosylation that need to be addressed before translation between the two can be made:

3.1 - Supplemental Figure 1 shows a marked upregulation in St6Gal2 upon TNFa stimulation in human cells, whereas St6Gal1 remains unaffected. However, in the mouse model the authors only examine St6Gal1 and do not detect St6Gal2 at all. Without overlap in expression between species, how can the authors claim that TNFa-induced St6Gal1 expression drives the same system in humans?

3.2 - Major differences are visible between the murine glycome (Supplemental Figure 3) and the human glycome (Supplemental Figure 4). These include N-glycolylneuraminic acid (high in mouse), alpha-linked galactosylation (high in mouse), and N-acetylglucosamine extensions (high in human). How do these distinct glycan epitopes factor into the analyses performed by the authors? Alpha-galactosylation, for example, may be a valid end-capping modification of murine glycans without altering the cells proinflammatory potential.

3.3 - Regarding the SNA and WGA assays, do N-acetylneuraminic acid and N-glycolylneuraminic acid have the same binding affinities? Could a lower lectin signal be explained by a switch from the one sialic acid type to the other?

3.4 - Mouse sialic acids are known to potentially carry acetyl groups, residues which are typically lost after permethylation analysis. Did the authors try analyzing the glycosylation without permethylation? Could acetylation prevent recognition of sialylation by SNA or WGA?

4 - While the authors have focused on N-glycosylation, some of the indicated glycosyltransferases (particularly GCNT1 and GCNT2) affect O-glycans as well, and these would equally be picked up by the current lectin analyses. Why did the authors focus on interpreting the lectin data by N-glycosylation alone, rather than by O-glycosylation or by the combination of the two

5 - None of the presented analytical methods informs on whether the glycans themselves are changed, or whether there is a different expression of glycoproteins carrying those glycans. To state that fibroblasts alter glycosylation quality by TNF α -induced ST6Gal1 expression, an experiment needs to be included showing the change on a single glycoprotein or glycosylation site. Fig 3, for example, shows large decreases in sialylation without corroborating increases in nonsialylated species. This suggests a change in glycoprotein expression rather than protein glycosylation. As it is, the reported decrease in sialylation could, for instance, also be explained by an influx of IgG (which primarily exhibits nonsialylated glycans) or something similar.

6 - A small comment, on page 5 "glycocalix" should be "glycocalyx".

Reviewer #3 (Remarks to the Author):

This manuscript presents a set of elegant experiments that were performed with the aim to determine the impact of inflammation/an inflammatory environment on the glycome of fibroblasts. The underlying hypothesis is that the composition of the glycome and, hence, the glycosylation of individual glycoproteins, could significantly impact on fibroblast function, which could be of relevance to, in this case, rheumatoid arthritis. By combining flow-cytometric (lectin-binding) assays with transcriptomic analyses and MS-based glycoanalytics, the authors demonstrate that inflammatory environments indeed associate with differential fibroblast glycomes and that differential fibroblast glycosylation is found in synovial fibroblasts (SF) isolated from arthritic joints compared to fibroblasts isolated from non-inflammatory tissues/compartments. Using murine SF, the authors further demonstrate a significant reduction of alpha2,6-linked sialic acid content in SF which relates to a TNF- α induced downregulation of ST6GAL1. This observation seems confined to the CD90+ subset of SF, which is interesting. Finally, a smaller set of data is provided that suggests that the reduced overall, alpha2,6-linked sialylation of SF might allow for enhanced SF responsiveness to galectin-3 stimulation, which would translate to a functional consequence of potential pathophysiological relevance.

The research question is very timely in the context of current/recent advances in the field in dissecting/understanding SF composition, the chosen approach uses and combines state-of-the-art technology, the work is extensive and the data are clearly and well presented. My main concern is that the authors base a large body of their conclusions on data obtained from murine fibroblasts, with the murine data not convincingly matching the human data. The work is detailed, in-depth and cohesive as to the effect of TNF- α on murine SF sialylation and sialyltransferase expression and has the interesting aspect of SF subsetting, all of which needs to be acknowledged.

The authors correctly claim that they provide an “extensive description of the SF glycome in health and disease”. The manuscript comes short, however, in demonstrating the functional relevance of the reduced sialylation, as the galectin-3 part is far less convincing. In fact, the manuscript title claims that the “Loss of α 2-6 sialylation promotes the transformation of synovial fibroblasts into a pro-inflammatory phenotype”, which is a conclusion that is not sufficiently supported by the data, especially as it suggests that this is the case in human RA. As such, this is very reminiscent of a large body of work that focuses on the pro-inflammatory effector functions conferred to IgG by a lack of sialic acid on Fc-glycans, which is convincing in the mouse but much less so in the human.

In detail:

1. The lectin-binding/flow cytometry data in Figure 1 (human), in particular those for SNA, do not fit the concept presented later on (mouse), as one would expect reduced SNA-binding by RA-derived SF (i.e. inflammatory) vs. dermal fibroblasts (i.e. non-inflammatory). The data show the opposite and the authors should explain/comment on this.

2. The dataset of Slowikowski (Suppl. Fig. 1 A) shows no effect of TNF α stimulation on ST6Gal1 expression by (human) fibroblasts, which contrasts the RNASeq findings presented later on using murine fibroblasts. In Suppl. Fig. 1 C, data on murine fibroblasts are presented, but primary human fibroblasts at this stage (e.g. human foreskin) could/should have been used.

3. In suppl. Fig 3 and 4, the authors describe that the human and mouse SF N-glycomes seem to be ‘well preserved’ based on a ‘similar structural profile’, but this is too optimistic and also not entirely correct, as rather large differences are depicted in the two suppl. figures, in particular with regard to the presence of many triantennary and higher order structures (present in the murine, absent in the human glycome) and the (known) absence of N-glycolylneuraminic acid in the human. To support the statement, a full comparison should be made, indicating the relative frequencies/abundances of the individual glycoforms, total sialylation/galactosylation/fucosylation etc, and the presence/absence of particular glycans and glycan structures.

4. Figure 7D is highly relevant as it is the only point at which an attempt is made to directly show the (potential) consequences of the reduced sialylation on fibroblast function. Presented are murine data and the response of SFs to galectin 3 stimulation. The authors highlight the differential responsiveness of LS-SFs (no response) versus HS-SFs (enhanced IL-6 secretion), but somehow omit to mention that naïve SFs also respond with enhanced IL-6 secretion, which is surprising, as one would expect that SFs from naïve, non-arthritic mice do not show a similar behavior as SFs derived from very inflamed paws. In fact, the gene expression pattern of the naïve and HS SFs are quite similar (7A), which does not fit the concept.

5. In line with the above, it seems an over interpretation of the data to state that ‘our conclusion of sialylation being an anti-inflammatory factor not only in the mouse model, but also in human RA.’ As the data are at the current stage, reduced sialylation is a marker of CD90+ SF activation (by TNF- α), but the functional link as suggested here and in the title is not convincing yet.

Minor comments:

- Fig. 4B: how many times was this experiment done/how many mice? How were the data quantified?

- the manuscript needs some proof reading

Response to reviewer's comments:

We would like to thank the reviewers for their comments and efforts towards improving our manuscript.

Reviewers raised concerns regarding the conclusions that sialylation is causally linked to pathogenic cell phenotypes and had questions about potential discrepancies between mouse and human biology. We agree that a stronger evidence of the functional consequences of reduction of sialylation would substantially improve our data. We have now conducted new experiments and analysis to address these concerns.

Our new data show that removal of sialic acid from SF glycoconjugates is sufficient to activate these cells in the absence of any other stimulation, inducing IL-6 and Ccl2 production. This demonstrates that loss of sialylation alone can trigger inflammatory pathways in SFs, supporting the conclusions presented in the manuscript. We have also added new experiments to compare mouse and human glycomes. Finally, we have clarified some areas of our manuscript that may have been misleading in the original submission.

In the following, we highlight these general concerns. Replies to other comments specific to each reviewer can be found after this.

General comments to all reviewers

#1 is sialylation causally linked to synovial fibroblast pathogenesis?

We have conducted experiments with targeted modification of the glycosylation. Our goal was to evaluate whether there is causality between reduced sialylation and SF-mediated inflammation. We decided to reduce SF sialic acid from healthy synovial fibroblasts to evaluate the potential impact on inflammatory responses. Two approaches were undertaken, i) silencing of ST6Gal1 expression using siRNA (Figure 9a&b) and ii) enzymatic removal of surface sialic acid using exogenous recombinant sialidase (Figure 9, c&d).

ST6Gal1 siRNA downregulated the expression of α 2-6 sialic acid (Assessed by qPCR and lectin SNA binding, Figure 9a). Once siRNA efficacy was confirmed, we evaluated the expression of pro-inflammatory IL-6, Ccl2 and MMP3 in cell culture supernatants, including AllStars siRNA (Qiagen®) as a negative control. Silencing of ST6Gal1 increased expression of IL-6 and Ccl2 ($p < 0.05$, $n=5$ experiments), with a small effect on MMP3 production (no statistical difference, $n=5$ experiments) (Figure 9b). This supports the hypothesis that high levels of α 2-6 sialic acid maintain cell homeostatic ability, whereas deficient expression of this molecule triggers inflammatory responses.

To corroborate this result, we used recombinant sialidase from *Clostridium perfringens*. This was an effective and quick method to remove surface sialic acid without interfering with gene expression. Incubation of cells with sialidase for 30' removed >50% of sialic acid from the cell surface. qPCR analysis showed that sialidase treatment *per se* induced a strong and consistent up-regulation of IL6 mRNA (4 fold) and Ccl2 mRNA (8 fold)[$n=5$ experiments], with no statistical significance in MMP3 mRNA expression (Figure 9c&d). These results agree with the siRNA experiments, indicating that the drop of sialic acid in SFs activates pathways to produce inflammatory IL-6 and Ccl2, with a more limited effect on MMP3 production. Additionally, we have evidence that TLR4-mediated responses are modulated by sialylation. Synovial fibroblasts with reduced sialylation (using siRNA and sialidase approaches) show an attenuated response to the TLR4 ligand LPS. We have not included this in the revised manuscript, but data are available to reviewers in response to reviewer 2, question 5 (Figure Q5-Reviewer 2). Overall, our new results prove that the content of sialic acid in SFs acts as a cellular signal to either repress or trigger cytokine production.

Results and discussion have been updated in the revised manuscript. We discuss how these new data strengthen the hypothesis that sialylation is modulating inflammatory activity of CD90+ sublining fibroblasts. IL-6 and Ccl2 are effector molecules associated to CD90+ sublining fibroblasts, whereas MMP3 is linked to CD90- lining subsets (Mizoguchi, Nat. Comms 2018). Our *in vivo* observations show that CD90+ fibroblasts are hyposialylated during experimental arthritis, and exogenous removal of sialic acid activates responses attributed to CD90+ sublining fibroblasts.

#2 is there a discrepancy between mouse and human data?

We feel that we have inadvertently misled the reviewers. More specifically, we refer to the results presented in Figure 1 and Supplemental Figure 1 in the original manuscript. Reviewers indicate that these results are in discrepancy with the mouse data. However, we did not intend to compare these data with the mouse model. We would like to highlight that these human cells were isolated from patients undergoing joint replacement surgery. This procedure is the last choice for patients that have experienced chronic joint inflammation for extended periods of time, often in the range of decades. They are also patients that have been treated for years with strong immunosuppressive drugs, like biologics targeting cytokines such as TNF α or IL-6. Given that our aim was to investigate the effect of cytokines on SF-glycosylation and inflammation, these results cannot reveal basic biological mechanisms and should not be compared with the animal model. In fact, after initial experiments with joint replacement samples, we decided to use the mouse model because it allowed us to investigate mechanisms associated to disease initiation without the interference of immunosuppressive drugs or over-extended chronic inflammation.

Nevertheless, these data were still relevant for us. They represent preliminary data that supported our working hypothesis: the cytokine milieu found in the inflamed joint determines distinct SF-glycosylation. Thereby, SFs isolated from different anatomical locations or different conditions (like OA and RA) would have distinct glycosylation signatures reflecting unique microenvironments. That was the observed experimental result, which provided the scientific and ethical reassurance to conduct further experiments in animal models. Ultimately, the animal model represents a more appropriate tool to describe basic mechanisms linked to health and disease, especially in an area of research where very little was known. To avoid any further misunderstanding, we have clarified this and we present these human results as supplementary data. We hope that the revised version of the manuscript delivers now a more clear message and discussion.

The main body of work is based on the murine model of disease, but we agree with the reviewers that it is important to determine whether our findings could be translated to the human disease. Major differences between mouse and human SF glycome could compromise future translational research. We had shown MS-based data with the annotated glycome for murine SFs from healthy synovium and cells from one RA patient. This was not totally conclusive and perhaps we did not have enough experimental evidence to claim that human and murine SF glycome were comparable. We expanded these experiments with a larger number of samples and a more systematic analysis. In this revised version, we show a detailed glycomic analysis for 4 murine samples from healthy animals, and human samples from 3 independent osteoarthritis (OA) patients. The ideal tissue would be healthy human SFs, but this is very rare in clinic and it would be difficult to get the amount of biological material required for MS experiments. Thus, as a form of Arthritis with a usually reduced inflammatory component, OA SFs are still a suitable candidate. In fact, it is common to find this approach in the literature. Mean \pm SD of relative expression for each N-glycan structure is now shown in murine and human cells (Supp. Figure 4&5). N-glycans comprising 75% and 90% of the total expression have been highlighted to show a more comprehensible visualization of the data. We also grouped all the N-glycans according to common structural themes and we compared total expression of these groups in murine and human cells (Supplemental Figure 6). The selected groups are as follows: 1- high mannose, 2- Core fucosylated, 3- Sialylated, 4- Sialylated bi-antennary, 5- non-sialylated bi-antennary, 6- glycans with 3 LacNAc motifs, 7- glycans with more than 3 LacNAc motifs, 8- hybrid, 9- bi-secting GlcNAc, 10- terminal fucosylated. Murine and human glycome are more similar than anticipated, aside from the inherent interspecies differences (terminal alpha-galactosylation and N-glycolylneuraminic are not synthesized in humans). The highest expression was in both cases the high mannose glycans, and differences in the rest of the groups were non-significant. The only significant difference was observed in the relative expression of non-sialylated bi-antennary N-glycans, which is probably a consequence of the lack of alpha-galactosylation in human cells (terminal galactosylation may prevent further extension of the antennae in mice). However, these more complex structures represent less than 10% of the total glycome. Finally, total sialylated N-glycans were expressed at similar levels in mouse and human.

Additionally, we evaluated the expression of ST6Gal1 in human OA synovial fibroblasts upon TNF α stimulation (Figure 5d). The TNF α -ST6Gal1 link is the key finding from the mouse model, and it was reproduced in the human cells. TNF α down-regulated ST6Gal1 mRNA expression, corroborating the murine data and supporting the human data shown in figure 8 from active naïve and remission patients.

We have provided initial evidence to support the translational potential of our findings, but further work with human tissue will be needed to evaluate the impact of sialylation on SF-mediated pathology. This future work will consider different phenotypes and disease stages, something that we will address in follow-up studies.

In the following sections, we respond to the comments that were more specific to each referee.

REVIEWER COMMENTS

Reviewer #1 (Remarks to the Author):

In this paper Pineda and colleagues explore sialylation of RA and CIA synovial fibroblasts. TNF is held responsible for alpha-2-6 hyposialylation of CD90+ sublining FLS, and this may be a marker, and perhaps even a cause for aggressive behavior of the FLS. This work is interesting but complex and occasionally paradoxical or puzzling. I have a few specific comments.

We thank the reviewer for reading the manuscript and the constructive feedback. The referee's comments about functional experiments fall under the umbrella of general comments and have been addressed in detail in the first section of our general comments.

The other comments and questions include:

Q1- Why are the changes in sialylation confined to one subset of FLS?

This is a very relevant question, not only regarding the glycobiology of the stromal compartment in RA, but in the entire disease pathogenesis. We acknowledge the importance of the question, and we feel that a conclusive and complete answer to it will require a long-term collective effort. This is in fact where our research is moving towards, along with deeper collaborations with clinical research groups. We have collated our results and some published work to speculate about potential mechanisms. Recent single cell RNA-Seq studies have identified the sub-lining CD90+ FLS as the stromal population driving disease pathogenesis in RA (Zhang F, Nat Immunol. 2019; Mizoguchi F, Nat Commun. 2018). However,

despite the latest technological advances we still do not fully understand the molecular mechanisms. Similarly, we cannot define the pathways that render CD90+ FLS hyposialylated at this stage. The distinct synovial environment where sublining fibroblasts sit might play a role, along with genetic and epigenetic factors that could condition the response to inflammatory cytokines. Because TNF α induces a strong down-regulation of ST6Gal1/sialylation, it would be possible that elements of the TNF α signaling pathways are different in lining and sublining fibroblasts. We have used the Accelerating Medicines Partnership (AMP) Rheumatoid Arthritis (RA) project (<https://immunogenomics.io/amp/ra/>) to test this hypothesis. This public dataset provides single cell RNA-Seq data in four distinct fibroblast subsets in RA synovium. The figure below shows the TNF α signaling pathway (KEEG database) and the relative expression of TNF α receptors and associated signaling molecules in each fibroblast subset. Sublining CD90+ subsets (SC-F1, SC-F2 and SC-F3) express significantly higher TNFR2 expression than lining CD90- cells (SC-F4), whilst all subsets showed similar expression of TNFR1. Interestingly, it has been shown that IL-6 production is decreased following TNFR2 blockade in RA FLS (Ma Z, Cytokine. 2016) and IL-6 is more expressed by CD90+ FLS (Mizoguchi F, Nat Commun. 2018). This suggests that there can be a correlation between TNF receptor expression and lining and sublining FLS. Moreover, individual subsets show differential expression of key signaling molecules, such as TRADD, FADD and members of the TNF receptor associated factors (TRAF) family. Such differences between subsets could trigger very different responses upon TNF α stimulation, including changes in glycosylation.

We are aware that these are preliminary observations and further work needs to be done. We are currently working on that direction. We do agree with the reviewer that this is a key question in the field and it needs to be addressed in detail. We have expanded this idea in the discussion of the revised manuscript.

Cluster annotations

Fibroblasts (CD45⁻ Podoplanin⁺)

- SC-F1: CD34⁺ sublining fibroblasts
- SC-F2: HLA⁺ sublining fibroblasts
- SC-F3: DKK3⁺ sublining fibroblasts
- SC-F4: CD55⁺ lining fibroblasts

Q2- To what extent do the glycomic patterns from the FLS signify changes in glycolipids versus glycoproteins?

This is an interesting question, although we feel that this comparison is beyond the scope of our research. We do not have the data or expertise to conduct these experiments now, but it is something that we would be interested to study in the future. We cannot rule out any effect on glycolipids, and we don't know whether glycolipids follow a similar pattern that that observed in glycoproteins. However, our available data suggest that the glycan moiety of glycolipids may have a more limited role. We have searched in our RNA-Seq data for expression of enzymes involved in ganglioside synthesis (see table below). Hierarchical comparative expression between naïve and arthritic cells showed no correlation between groups, as shown in the heatmap (below), suggesting that their expression is not linked to disease pathology. There are few published lipidomic studies in the context of RA. Kosinska et al reported differential expression of glycolipids in synovial fluid from healthy, OA and RA patients, including sphingolipids and glycerophospholipids (Kosinska et al, PLoS ONE, 2014). In that study, hexosylceramides and lactosylceramides structures are analysed, and no significant differences were observed. On the other hand, the sialic acid-containing Ganglioside GM3 has been reported to be down-regulated in RA and deletion of the GM3 synthase gene (ST3Gal5) exacerbated inflammation in the CIA model (Tsukuda Y, PLoS One. 2012). ST3Gal5 was found in our murine fibroblasts (see heatmap below) but its expression was not differentially regulated in arthritic mice. Further work is therefore required to allow meaningful comparisons between glycoproteomics and glycolipidomics.

Gene	Common name	Main acceptor
UGCG	GlcCer synthase	Ceramide
B4GALT6	LacCer synthase	Glucosylceramide
ST3GAL5	GM3 synthase	Lactosylceramide
ST8SIA1	GD3 synthase	GM3, GD3
ST8SIA5	GT3 synthase	GD3, GM1b, GD1a, GT1b
B4GALNACT1	GM2/GD2 synthase	GA3, GM3, GD3, GT3
B3GALT4	GM1a/GD1b synthase	GA2, GM2, GD2, GT2
ST3GAL2	ST3Gal II	Galβ1-3GalNAc-R
ST6GALNAC 3	ST6GalNAc III	Neu5Aca2-3Galβ1-3GalNAc-R
ST6GALNAC5	ST6GalNAc V	Neu5Aca2-3Galβ1-3GalNAc-R

(R = LacCer, GM3, GD3, or GT3).

Q3- The range of functional studies of the FLS in this paper is rather limited. Prior observations from other groups do not suggest that IL-6 hyper-production and proliferation/migration/cartilage invasion are performed by the same FLS subsets.

Please see our response to the common comments in the first section of this document.

The functional relevance of SF sialylation is now demonstrated in the new set of experiments, as removal of sialic acid induces IL-6 and Ccl2 expression (Figure 9). Interestingly, desialylated cells do not increase MMP3 expression, suggesting that only specific pathways are dependent on α2-6-sialic acid. This inflammatory signature (IL6^{high}, Ccl2^{high}, MMP3^{low}) matches the gene expression pattern of CD34+ CD90+ sublining fibroblasts (Mizoguchi F, Nat Commun. 2018), in agreement with our data showing reduced ST6Gal1 expression in CD90+ fibroblasts. The reviewer is right about IL-6 production and migration/proliferation genes being associated to distinct subsets. MMP3 is very highly expressed by CD90-lining fibroblasts, but sublining CD90+CD34- co-express MMP and secreted cytokines, like IL-6 (Mizoguchi F, Nat Commun. 2018). Our data in figure 6 show cells sorted based on the expression of CD90 only, which includes all sublining cells, explaining the co-expression of IL-6 and MMP3.

Q4- Are the RA FLS lines developed for this study (from patients with active disease) CD90+ and exclusively representative of the sublining, hyposialylated FLS?

RA FLS cells are expanded populations from the whole synovium and cells are not exclusively representative from any anatomical location. Nevertheless, most of the cells in these experiments are CD90+ as they are the dominant subtype, but they still represent the heterogeneity found in the original tissue. Now, we plan to use the basic mechanisms described here to conduct an exhaustive study in human RA, where we will evaluate the role of sialosides in different disease stages and specific cell subsets.

Q5- PHA also binds strongly to T lymphocytes. In the tissue sections how is lymphocyte binding distinguished from FLS binding?

Vimentin staining is included to identify stromal cells. Vimentin is restricted to cells of mesenchymal origin, which allows identification of fibroblasts in relation with their anatomical location. Vimentin- PHA+ staining represents therefore immune cells recognized by PHA, although we cannot claim that they are lymphocytes. We have now added some signs in the images to facilitate visualization of fibroblast and PHA staining.

Q6- The writing is uneven, Some of the sections are not well stitched together. For example, IL-22 appears out of nowhere and then disappears. Irritating phrases such as "an under-sialylated and highly pro-inflammatory microenvironment that contributes to an amplificatory inflammatory network that perpetuates chronic inflammation" are encountered, much to the displeasure of the reader.

We apologize for these oversights, thank you for pointing this out. We have revised the text.

Regarding IL-22, these data were included to provide proof of concept to support the idea that cytokine signaling can modulate glycosylation patterns. We chose IL-22 in combination with other stimuli (IL-17, IL-6, LPS and IFN β) because IL-22 has been described to exert both pro- and anti-inflammatory actions depending on the context. Our main goal was to provide an adequate context to readers that know enough about inflammation but are less familiar with the role of glycans in this process. We have now modified the text to clarify that. Perhaps we have involuntarily distracted the reader with these data, that are not directly related to our main findings. We could eliminate these data from the final version of the manuscript if the reviewer thinks that this will make a more comprehensive text.

Reviewer #2 (Remarks to the Author):

Wang Y and coauthors report the in-depth investigation into the glycosylation differences between various fibroblasts from patients with rheumatoid arthritis or osteoarthritis, as well as those from an approximate mouse model (collagen-induced arthritis) in which cytokine treatment could be performed. Glycosylation was determined by lectin interaction and mass spectrometry and the authors coupled these to transcript analysis and cytokine intervention, ultimately arriving at the conclusion that alpha-2,6-linked sialylation is decreased in fibroblasts from an arthritic background.

The presented body of work is impressive, as are the suggestions provided thereby. However, I do have several reservations about the interpretation of the lectin and glycomics experiments, as well as about eagerness with which biological conclusion are drawn. My specific comments can be found below.

We thank the reviewer for the positive comments and for the insightful review of our manuscript. These comments have not only helped us to improve the manuscript, but they have also led us to interesting questions for our future research.

Q1 - The manuscript title suggests that the loss of alpha-2,6-linked sialylation causes the development of proinflammatory synovial fibroblasts, but the data in the manuscript suggests only a correlation between the two at best. To claim causation the authors have to include experiments with targeted modification of the glycosylation.

We have performed experiments with targeted modification of the sialylation as suggested by the reviewer. Please note that we explained these under the set of responses common to all three referees.

Our new results show that eliminating sialic acid is sufficient to induce IL-6 and Ccl2 production, which establishes causation between the observed glycosylation changes and synovial fibroblast activation. This now opens interesting questions that we would like to address in the future, such as the role that siglecs and galectins play in synovial fibroblast activation.

Q2 - Figure 1 does not generally support the conclusions of the authors as no apparent distinction can be seen in WGA and SNA staining between fibroblast locations. Can the authors explain this discrepancy?

The human synovial fibroblasts in Figure 1 (original submission) are from patient cohorts that have been under biologic disease-modifying antirheumatic drugs for years and ultimately required joint replacement surgery. Results show some differences between glycosylation patterns and disease types or anatomical locations, but they are not appropriate to study early cytokine-dependent mechanism. As an example, most of these RA patients have been treated for years with anti-TNF α neutralizing antibodies. Thus, TNF α -dependent biology will necessarily be modified. Considering that, it is less surprising to find no distinction in WGA and SNA staining, possibly a reflection of altered TNF α -related actions. This has been discussed in the general response to all reviewers and added to the revised text. As we indicated before, we cannot compare the data in Figure 1 with the results generated with the mouse model. In fact, the nature of these human samples was the reason that made us use the experimental murine model. The CIA model allowed us to identify changes in the synovial fibroblast glycome that could be of relevance for initiation of human RA. Now our plans include more focused studies using defined patient cohorts, like those shown in Figure 8. Due to the heterogeneous pathophysiology of RA, this will require a higher number of samples, including different disease stages (initiation, perpetuation, remission).

Q3 - There are several profound differences between human and murine glycosylation that need to be addressed before translation between the two can be made:

Q3.1 - Supplemental Figure 1 shows a marked upregulation in ST6Gal2 upon TNF α stimulation in human cells, whereas ST6Gal1 remains unaffected. However, in the mouse model the authors only examine ST6Gal1 and do not detect ST6Gal2 at all. Without overlap in expression between species, how can the authors claim that TNF α -induced ST6Gal1 expression drives the same system in humans?

There are some important differences between data in Figure 1 (original manuscript) and the mouse data. First, and perhaps the most important, the nature of the human RA samples used in the study, something that we have already discussed. Second, human fibroblasts in Figure 1 were stimulated with 1 ng/ml TNF α , and we used 10 ng/ml.

We have conducted some in vitro experiments to see whether the TNF α -ST6Gal1 link exists in humans too. We expanded human OA SFs, as a less inflammatory model for the human system. We have shown that these fibroblasts and murine naïve cells express comparable glycomes (Supplemental Figures 4-6, see comment below). We treated these human cells with TNF α to evaluate ST6Gal1 expression, results that are now included in the revised manuscript (Figure 5d). ST6Gal1 mRNA was significantly reduced in response to TNF α , corroborating the data from the mouse model.

We could not detect expression of ST6Gal2 in the mouse fibroblast, but the reviewer is right noting that this enzyme was present in the RNASeq human dataset generated by Slowikowski et al. We evaluated expression of ST6Gal2 in human OA SFs (Figure 5d). Although we did not expect to detect ST6Gal2, human SFs expressed this glycosyltransferase, albeit the expression levels were lower than ST6Gal1. Interestingly, TNF α also reduced ST6Gal2 expression, indicating that TNF α can down-regulate total α 2-6 sialylation in humans. This is interesting and it will be something to consider in future studies. This result is now shown in Figure 5d and discussed in the manuscript.

Q3.2 - Major differences are visible between the murine glycome (Supplemental Figure 3) and the human glycome (Supplemental Figure 4). These include N-glycolylneuraminic acid (high in mouse), alpha-linked galactosylation (high in mouse), and N-acetylglucosamine extensions (high in human). How do these distinct glycan epitopes factor into the analyses performed by the authors? Alpha-galactosylation, for example, may be a valid end-capping modification of murine glycans without altering the cells proinflammatory potential.

This question is addressed in the first section of this document, please see that reply.

There are some specific points about this that we address here:

Although some N-glycolylneuraminic (Neu5G) can be incorporated in human cells from diet or culture medium, Neu5G is not synthesized by human cells. Similarly, human cells do not synthesize alpha-linked galactosylation. Therefore, these differences will be always part of animal models, not only in this study but in any experiment using rodents as models. In our model of arthritic SFs, the relative expression of sialylated N-glycans (including Neu5Ac + Neu5Gc) is not significantly different in mouse and human (mouse $19.3 \pm 6.5\%$, human 26.2 ± 10.8) (Supplemental Figure 6). Relative expression of bi-antennary sialylated glycans [the most abundant type in mouse and human after high-mannose] is even closer (mouse $12.7 \pm 3.17\%$, human 12.5 ± 4.2). Despite their structural difference, Neu5Gc and Neu5Ac still share most of the characteristics that define the family of sialic acids (nine carbon acidic sugars, negatively charged, binding to specific carbohydrate binding proteins). Importantly, murine alpha galactosylation does not seem to affect the relative levels of sialylated glycans comparing mouse and human, and the major impact seems to be in LacNAc extensions. We also see that these more complex N-glycans represent less than 10% of the total glycome in both mouse and human. Although we acknowledge that this will add some interspecies variability, the mouse model appears to be an appropriate model to study the synovial fibroblast glycome, accepting the inherent differences associated to humans and other mammals. We have now discussed this in the revised version of the manuscript. The new analysis shown in Supplemental figure 6 provides now a more comprehensive approach to compare structural differences and similarities between mouse and human.

Q3.3 - Regarding the SNA and WGA assays, do N-acetylneuraminic acid and N-glycolylneuraminic acid have the same binding affinities? Could a lower lectin signal be explained by a switch from the one sialic acid type to the other?

Song et al (Song X., J Biol Chem. 2011) developed a chemoenzymatic method to synthesize and print a wide selection of sialosides on glycan microarrays. SNA and MAA lectins were tested in that study, showing specific lectin binding for Neu5Ac and Neu5Gc containing glycans. A full report can be seen in the original paper. We have selected some examples from this work (see figure below) to answer the reviewer's question. Song's manuscript indicates that SNA display similar affinity towards both forms of sialic acid, whereas MAA may have an increased affinity for Neu5Ac. Thereby, MAA binding could be affected by different ratios of Neu5Ac/Neu5Gc, although we did not observe any significant difference using this lectin.

To confirm that our observations were due to changes in global sialylation and not a switch in Neu5Ac to Neu5Gc, we calculated the ratio of the Neu5Gc:Neu5Ac most abundant forms in our glycomic data set, comparing Naïve versus arthritic mice. Results (shown below) show that there are no significant changes in the ratio Neu5Gc:Neu5Ac between healthy and arthritic conditions. This has been commented in the revised manuscript. We cannot rule out that WGA can present different affinity towards Neu5Gc or Neu5Ac, but we have not detected any Neu5Ac/Neu5Gc variations and WGA has not been used for mechanistic studies in the animal model.

Q3.4 - Mouse sialic acids are known to potentially carry acetyl groups, residues which are typically lost after permethylation analysis. Did the authors try analyzing the glycosylation without permethylation? Could acetylation prevent recognition of sialylation by SNA or WGA?

Thanks for pointing this out. O-acetyl esters at C4, C7, C8 or C9 of sialic acids are very common modifications that can affect immunological processes. Data shown in the aforementioned paper (Song X, J Biol Chem. 2011) indicate that acetylation of both Neu5Ac and Neu5Gc resulted in slightly increased binding of SNA. Effects of acetylation on MAA binding were less consistent, and it varied significantly depending on the main structure that the sialic acid was bound too. We have summarized some of these data extracted from Song's paper below for SNA and MAA binding, selecting forms of acetylated N-glycans expressed in mammalian cells. Again, we have less information regarding WGA, but this lectin was not used to evaluate changes in experimental arthritis.

We have not analysed the glycome in non-permethylated samples. We feel that studying modifications of the sialic acids will add an extra layer of complexity. This will require additional expertise and time to define the sialylome to that level of molecular detail. This is something that we will be looking at in the future since it may be relevant to understand interactions of synovial fibroblasts with other immune cells. For example, 9-O-acetylation on B-cell surface sialic acid blocks recognition by α 2-6 sialic acid-binding immunoglobulin-type lectin Siglec-2 (CD22), inhibiting receptor signaling and B-cell activation (Cariappa A, J Exp Med. 2009). Our hypothesis is that TNF α -dependent down-regulation of α 2-6 sialic acid stops homeostatic Siglec-2 signalling, allowing B cell activation by SFs. Based on this reviewer's comment, we will also consider differential acetylation as a further mechanism in the perpetuation of local inflammation.

To get some preliminary data, we have conducted some analysis of the RNA-Seq presented in the manuscript. The presence and levels of O-acetylation result from combined functions of sialic acid O-acetyl transferases (SOAT) and esterases (SOAE), some of which have been identified and functionally demonstrated, while others are not yet characterized. We have looked at the expression of *Casd1* and *Siae* genes in murine synovial fibroblasts, both in healthy and arthritic mice (See data below), as described SOAT and SOAE genes. Interestingly, both genes were expressed, suggesting that sialic acid can be acetylated. However, no significant difference was observed between healthy and CIA cells.

This data suggest that acetylation is present in the system, although it may not be significantly regulated during disease. However, we cannot rule out that a differential acetylation of sialic acid can affect sialic acid-dependent inflammation. We have now included the relative expression of *Casd1* and *Siae* genes in Fig 5a to provide further description of sialic acid biosynthetic pathways. We have also included the *CMAH* gene responsible of converting CMP-Neu5Ac is to CMP-Neu5Gc in murine cells. This offers a more descriptive and complete analysis. The potential role of sialic acid modifications, like acetylation, has been added to the manuscript.

Q4 - While the authors have focused on N-glycosylation, some of the indicated glycosyltransferases (particularly GCNT1 and GCNT2) affect O-glycans as well, and these would equally be picked up by the current lectin analyses. Why did the authors focus on interpreting the lectin data by N-glycosylation alone, rather than by O-glycosylation or by the combination of the two.

O-glycans are certainly picked up by lectin analysis and may have a significant effect on cell biology and cell-cell communication. We initially focused on N-glycosylation for technical reasons. N-glycans can be removed from proteins by enzymatic hydrolysis, using purified N-glycosidase as described in the manuscript. This provides a clean a reproducible method to isolate glycans, which made it appropriate for a semiquantitative analysis as shown in Figure 2. This was important for us as we needed a powerful and reliable method to minimize experimental error. Isolation of O-glycans on the other hand, requires a chemical approach, reductive elimination using KBH4. This adds more variability between experiments and increases the noise, making it less suitable for the type of analysis that we wanted to conduct.

Nonetheless, we agree with the reviewer and we feel that we cannot neglect O-glycosylation if we aim to provide a systematic study. For that reason, we have isolated O-glycans from healthy and arthritic synovial fibroblasts. O-glycans were chemically cleaved from peptides by reductive elimination and permethylated as described for N-glycans prior to MS-based analysis. These results are shown in the revised manuscript (Figure 3). The synovial fibroblast O-glycome was dominated by core-1 and core-2 structures, with high presence of sialic acids bound to galactose and an absence of fucosylation. This is in

line with the terminal modifications described in the N-glycome. Likewise, sialylated glycoforms were reduced in arthritic fibroblasts, along with an increase of non-sialylated core-1 and core-2 glycans, corroborating our results. Additionally, the ratio of intensity between m/z 925: 895 was assessed in healthy and arthritic fibroblasts to check the relative expression of Neu5Gc versus Neu5Ac. These two glycans are the most abundant sialylated structures in the O-glycome and no major difference was observed. This corroborates the observations discussed in Q3.3, confirming that there is no shift between sialic forms in disease.

Q5 - None of the presented analytical methods informs on whether the glycans themselves are changed, or whether there is a different expression of glycoproteins carrying those glycans.

To state that fibroblasts alter glycosylation quality by TNF α -induced St6Gal1 expression, an experiment needs to be included showing the change on a single glycoprotein or glycosylation site. Fig 3, for example, shows large decreases in sialylation without corroborating increases in nonsialylated species. This suggests a change in glycoprotein expression rather than protein glycosylation. As it is, the reported decrease in sialylation could, for instance, also be explained by an influx of IgG (which primarily exhibits nonsialylated glycans) or something similar.

Please let us reply to this comment in three different but related lines:

1- None of the presented analytical methods informs on whether the glycans themselves are changed, or whether there is a different expression of glycoproteins carrying those glycans [...]. Fig 3, for example, shows large decreases in sialylation without corroborating increases in nonsialylated species. This suggests a change in glycoprotein expression rather than protein glycosylation

None of the methods can demonstrate changes in glycosylation when used individually, but all of them together strongly support specific glycan remodeling rather than changes in the expression of glycoproteins. Lectin-based assays are the most general approach, and results could be a consequence of changes in glycan biosynthesis and/or protein expression. However, we also provide extensive description of the transcriptome, RNA-Seq and qPCR data, and MS-based glycomics, comparing naïve versus arthritic mice and non-stimulated versus TNF α stimulated cells. All approaches indicate a reduction in sialylation.

In this regard, we disagree with the reviewer's comment "Fig 3, for example, shows large decreases in sialylation without corroborating increases in nonsialylated species. This suggests a change in glycoprotein expression rather than protein glycosylation". In figure 3, there are structures that are significantly up-regulated in arthritic cells, glycan 2110 in cluster 2285 and glycan 2244 in cluster 2069. These two glycans are very abundant, 2244 alone counts for 2-3% of the total. This is a very high proportion, especially if we exclude the high-mannose glycans that represent >50%. The high increase in the relative amount of 2244 is very relevant because is the non-sialylated core for the glycans that show the highest decrease in sialic acid, those grouped in cluster 3026. Furthermore, non-fucosylated sialosides are less abundant, and indeed the accumulation of non-fucosylated bi-antennary core (2069) is less affected. Overall, the glycomics data show a reduction in sialylated accompanied with an increase in non-sialylated analogs.

Additionally, glycomics data show a distinct effect on sialic acid expression, without affecting other glycan modifications. We have calculated ratios in the glycome related to single glycan modifications, and sialylation is the only one that shows a change in tendency. If our results were a consequence of modulation of protein expression, we would affect all glycosylation pathways, but only sialic acid is changed (see table below, showing ratios of related modifications in healthy and CIA fibroblasts).

	Sia/Non-Sial	Fuc/Non-Fuc	High Man/Non-High Man	aGal/Non-aGal
Naïve	1.16	3.57	0.94	0.51
CIA	0.81	4.13	0.94	0.44

Sialic acid reduction can only be explained by modulation of glycosyltransferase activity, as observed at the transcriptomic level for ST6Gal1. Indeed, transcriptomics and glycan expression analysis (lectin-based and MS-based experiments) converge to support the regulation of glycosylation pathways as the cause for reduced sialylation. Validating this, our new data show that exogenous depletion of sialic acid induces the inflammatory phenotype observed in vivo and in vitro.

2- To state that fibroblasts alter glycosylation quality by TNF α -induced St6Gal1 expression, an experiment needs to be included showing the change on a single glycoprotein or glycosylation site.

Identification of the proteins that are modulated by sialylation is a very relevant question and it is indeed another key area of research for us in the future. However, we feel that this is beyond the scope of this manuscript in establishing solid functional glycomic foundations in synovial fibroblast biology. Of course, we fully agree with the reviewer, and we will use our glycomic data as a springboard to generate the required funds for further studies focused on glycoproteomics.

Nevertheless, we have tried to identify some of the molecules that may be sialylated on synovial fibroblasts, being of relevance for cell activation in RA. Based on the current literature, we think that Toll-like receptor (TLR) signaling and integrin-mediated migration are strong candidates. Both TLRs and fibronectin have been described as key regulators of

synovial fibroblast pathophysiology and they both are heavily glycosylated and sialylated molecules. To corroborate this hypothesis, we conducted some experiments to evaluate the sialylation of TLR4. Naive synovial fibroblasts isolated from healthy mice express α 2-6 sialylated TLR4 on the cell surface. TNF α stimulation reduced α 2-6-sialylated TLR4, but keeping high levels of non-sialylated TLR4. This experiment shows a single glycoprotein with modified sialylation. Supporting this, response to LPS (TLR4 ligand) was attenuated when sialylation was down-regulated via ST6Gal1 siRNA or sialidase treatment. These results are shown below (Figure Q5), providing a single pathway affected by sialic acid and further evidence for the functional relevance of sialic acid in SF-dependent inflammation. These results are not part of the revised manuscript, but can be included if it was necessary.

Figure Q5-Reviewer 2. Sialic acid modulates TLR4-mediated signaling in mouse SFs. A) Murine synovial fibroblasts were stimulated with TNF α for 12 and 24 hours. TLR4 was detected with a goat anti-mouse TLR4 antibody (Red, Santa Cruz Biot.) and α 2-6 sialic acid was detected with SNA (Green). B) IL-6, Ccl2 and MMP3 in cell culture medium upon LPS stimulation (12 hours, 100 ng/ml), ELISA results. C) fibroblasts were treated with siRNA for ST6Gal1 or Allstars control for 3 days, medium was refreshed and cells were stimulated with LPS or DMEM for 12 hours. Cytokine concentration was measured by ELISA. D) Sialic acid was removed from synovial fibroblasts using *C. perfringens* sialidase (CP; 30', 37 C). Fresh medium was added in the presence or absence of LPS as indicated and RNA was extracted after 3 hours. Expression of IL-6, Ccl2 and MMP3 was evaluated by qPCR.

3- As it is, the reported decrease in sialylation could, for instance, also be explained by an influx of IgG (which primarily exhibits nonsialylated glycans) or something similar.

In vivo experiments were conducted with sorted cells, and ex vivo experiments were conducted with expanded synovial fibroblasts. In all cases, cells were isolated, and either used or expanded in the absence of other cell types. We did not detect any other cell type in our experiments other than podoplanin and vimentin positive fibroblasts. An influx of IgG is possible in our experimental setup.

Q6 - A small comment, on page 5 "glycocalix" should be "glycocalyx".

Thanks.

Reviewer #3 (Remarks to the Author):

This manuscript presents a set of elegant experiments that were performed with the aim to determine the impact of inflammation/an inflammatory environment on the glycome of fibroblasts. The underlying hypothesis is that the composition of the glycome and, hence, the glycosylation of individual glycoproteins, could significantly impact on fibroblast function, which could be of relevance to, in this case, rheumatoid arthritis. By combining flow-cytometric (lectin-binding) assays with transcriptomic analyses and MS-based glycoanalytics, the authors demonstrate that inflammatory environments indeed associate with differential fibroblast glycomes and that differential fibroblast glycosylation is found in synovial fibroblasts (SF) isolated from arthritic joints compared to fibroblasts isolated from non-inflammatory tissues/compartments. Using murine SF, the authors further demonstrate a significant reduction of alpha2,6-linked sialic acid content in SF which relates to a TNF-alpha induced downregulation of ST6GAL1. This observation seems confined to the CD90+ subset of SF, which is interesting. Finally, a smaller set of data is provided that suggests that the reduced overall, alpha2,6-linked sialylation of SF might allow for enhanced SF responsiveness to galectin-3 stimulation, which would translate to a functional consequence of potential pathophysiological relevance.

The research question is very timely in the context of current/recent advances in the field in dissecting/understanding SF composition, the chosen approach uses and combines state-of-the-art technology, the work is extensive and the data are clearly and well presented. My main concern is that the authors base a large body of their conclusions on data obtained from murine fibroblasts, with the murine data not convincingly matching the human data. The work is detailed, in-depth and cohesive as to the effect of TNF-alpha on murine SF sialylation and sialyltransferase expression and has the interesting aspect of SF subsetting, all of which needs to be acknowledged. The authors correctly claim that they provide an "extensive description of the SF glycome in health and disease". The manuscript comes short, however, in demonstrating the functional relevance of the reduced sialylation, as the galectin-3 part is far less convincing. In fact, the manuscript title claims that the "Loss of a2-6 sialylation promotes the transformation of synovial fibroblasts into a pro-inflammatory phenotype", which is a conclusion that is not sufficiently supported by the data, especially as it suggests that this is the case in human RA. As such, this is very reminiscent of a large body of work that focuses on the pro-inflammatory effector functions conferred to IgG by a lack of sialic acid on Fc-glycans, which is convincing in the mouse but much less so in the human.

We thank the reviewer for the positive comments and constructive criticisms. Some of the comments have been already addressed in the reply to general comments, please see that section.

We acknowledge that the original manuscript lacked some functional experiments. In the revised version, we have conducted new experiments modifying the sialylation profile of synovial fibroblasts (Figure 9). We show now that removal of sialic acid is sufficient to induce IL-6 and Ccl2 secretion in healthy cells.

We also agree that further work was needed to validate these findings in the human disease. We have added new data to compare the mouse glycome with the human system in a more comprehensive way and we have clarified some of the aspects about the human data as explained before. Moreover, we have conducted some experiments showing that TNF α also down-regulate ST6Gal1 expression in human fibroblasts.

Overall, this provides initial evidence for potential translation of our findings in the murine model to the human system. Of course, we acknowledge that further work is still needed to understand glycosylation changes in human RA. This will be an important area of our future research. Therefore, we have corrected some of the interpretations that were wrongly attributed to human RA, referring them to the experimental model. We have removed the word rheumatoid from the title, presenting a more general approach to avoid any misinterpretation.

In detail:

Q1. The lectin-binding/flow cytometry data in Figure 1 (human), in particular those for SNA, do not fit the concept presented later on (mouse), as one would expect reduced SNA-binding by RA-derived SF (i.e. inflammatory) vs. dermal fibroblasts (i.e. non-inflammatory). The data show the opposite and the authors should explain/comment on this.

We have addressed this at the beginning of this document under the reply to general comments (Rebuttal, page 1, point #2). Please see also reply to question 2 from reviewer 2 (Rebuttal, page 5).

Q2. The dataset of Slowikowski (Supp. Fig. 1 A) shows no effect of TNF α stimulation on ST6Gal1 expression by (human) fibroblasts, which contrasts the RNASeq findings presented later on using murine fibroblasts. In Supp. Fig. 1 C, data on murine fibroblasts are presented, but primary human fibroblasts at this stage (e.g. human foreskin) could/should have been used.

Some of this has also been address under general comments (Rebuttal page 1, point #2). As explained, tissue from joint replacement surgery should not be directly compared to the results from the animal model. We hope that the revised version offers a better analysis now.

Please, see also reply to Q3 from reviewer 2 (Rebuttal, page 6). We have copied some of that reply here:

Reply to Q3R2: *“There are some important differences between data in Figure 1 (original manuscript) and the mouse data. First, and perhaps the most important, the nature of the human RA samples used in the study, something that we have already discussed. Second, human fibroblasts in Figure 1 were stimulated with 1 ng/ml TNF α , and we used 10 ng/ml. We have conducted some in vitro experiments to see whether the TNF α -ST6Gal1 pathway functions in humans too. We expanded human OA SFs, as a less inflammatory model for the human system. We have shown that these fibroblasts and murine naïve cells express comparable glycomes (Supplemental Figures 4-6, see comment below). We treated these human cells with TNF α to evaluate ST6Gal1 expression, results that are now included in the revised manuscript (Figure 5d). ST6Gal1 mRNA was significantly reduced in response to TNF α , corroborating the data from the mouse model.”*

We agree that the experiment shown in Sup. Figure 1C (Original submission) would benefit from human studies. However, we did not have access to human foreskin fibroblasts. Naïve mouse cells provide the appropriate context to demonstrate that cytokines can have a synergistic or antagonistic effect on glycosylation. Obtaining human fibroblasts suggested by the reviewer would be very challenging at the moment, and it would not substantially affect our main findings. However, we will consider conducting these experiments in the future.

Q3. In suppl. Fig 3 and 4, the authors describe that the human and mouse SF N-glycomes seem to be ‘well preserved’ based on a ‘similar structural profile’, but this is too optimistic and also not entirely correct, as rather large differences are depicted in the two suppl. figures, in particular with regard to the presence of many triantennary and higher order structures (present in the murine, absent in the human glycome) and the (known) absence of N-glycolylneuraminic acid in the human. To support the statement, a full comparison should be made, indicating the relative frequencies/abundances of the individual glycoforms, total sialylation/galactosylation/fucosylation etc, and the presence/absence of particular glycans and glycan structures.

Thanks for this comment. Again, this is also addressed under general comments.

We have conducted a more methodical study comparing the N-glycome of naïve healthy fibroblasts (n=4) with human fibroblasts isolated from OA patients (n=3). Relative intensity for each identified glycan has been calculated and shown in the figures, as explained in the reply to general comments. Relative intensity of distinct structurally related groups is shown in supplemental figure 6. Both human and mouse glycomes show very similar percentages of high mannose glycans, sialylated, bi-antennary sialylated and core fucosylated glycans. These groups constitute most of the glycome, with other structures (for example the tri- and tetra-antennary glycans being less than 10%). Likewise, there are no differences in the order of these structures if we consider relative expression, confirming the similarity between species. The major difference between mouse and human is linked to bi-antennary non-sialylated glycans and glycans with poly LacNAc units. This is probably a reflection of the murine alpha-galactosylation that is absent in humans. We have grouped more complex glycans as poly LacNAc containing structures, because based on the transcriptomic experiments, these forms are likely to contain multibranching structures in mouse and human, but the presence of other forms showing longer antennae cannot be completely resolved. Total sialylated glycans are similar in mouse and human, including N-glycolylneuraminic acid for the mouse.

Q4. Figure 7D is highly relevant as it is the only point at which an attempt is made to directly show the (potential) consequences of the reduced sialylation on fibroblast function. Presented are murine data and the response of SFs to galectin 3 stimulation. The authors highlight the differential responsiveness of LS-SFs (no response) versus HS-SFs (enhanced IL-6 secretion), but somehow omit to mention that naïve SFs also respond with enhanced IL-6 secretion, which is surprising, as one would expect that SFs from naïve, non-arthritic mice do not show a similar behavior as SFs derived from

very inflamed paws. In fact, the gene expression pattern of the naïve and HS SFs are quite similar (7A), which does not fit the concept.

We have added new functional experiments (Figure 9). The consequence of reduced sialylation is activation of pro-inflammatory cytokine synthesis, IL-6 and Ccl2. This confirms the functional role of sialylation on fibroblast function. Additionally, we generated data showing the effect of reduced sialylation on TLR4-mediated inflammation (see question 5, reviewer 2, page 10 of this document).

We apologize if we were not clear discussing the results of naïve and LS/HS fibroblasts. Galectin-3 can act as a danger-associated molecular pattern (DAMP), and it is not that surprising that naïve fibroblasts can respond to this danger signal. This would be part of immune mechanisms associated to galectins and acute responses. Naïve synovial fibroblasts are therefore equipped with the molecular tools to initiate acute responses upon galectin-3 binding. The question that remains unanswered in arthritis is why this inflammation persists. This could be explained by changes in fibroblast glycosylation. We propose that galectin-3 actions combined with changes in glycosylation can play a role in perpetuation of disease, as responses are exacerbated in fibroblasts from inflamed joints. This would be compatible with the ability of galectin-3 to initiate acute responses on naïve SFs in response to infections, for example. Sustained exposure to TNF α in chronic inflammation would reconfigure this otherwise temporal defense mechanism to an inflammatory loop.

There are important differences between naïve and HS fibroblasts. Non-sialylated structures are up-regulated, along with a strong up-regulation of a group of sialylated glycans (2822, 3415,3271,3037). It is also possible that differences between naïve and HS seem less obvious in this experiment, as the heatmap scores upon clustering are affected by the inclusion of the LS cells. Glycomic results comparing naïve versus inflammatory synovial fibroblasts only are extensively described in figure 2, including 4 experiments. Furthermore, it is possible that other structural aspects are affecting the ability of naïve, HS and LS fibroblast to respond to galectin-3. One factor could be the length or branching of the antennae found in the N-glycans. In fact, structures found upregulated in LS cells (cluster 3619) include >50% of additional LacNAc units. We don't know exactly the proportion of multiantennary versus extended antennae forms in our analysis, but it would be possible that both sialic acid and LacNAc extension vs branching contribute to modulate galectin-3 binding and responses. It also exist the possibility that LS cells are in a transition stage, in a preclinical stage of disease. Unfortunately, we cannot know what is the exact situation for these cells, circumstance that has been commented in the revised manuscript.

Q5. In line with the above, it seems an over interpretation of the data to state that 'our conclusion of sialylation being an anti-inflammatory factor not only in the mouse model, but also in human RA.' As the data are at the current stage, reduced sialylation is a marker of CD90+ SF activation (by TNF-alpha), but the functional link as suggested here and in the title is not convincing yet.

Our new data now show a direct link between sialylation and functional responses. We have revised the text to avoid over interpretations of the mouse data in relation with the human disease. We have also modified the title to 'Loss of a2-6 sialylation promotes the transformation of synovial fibroblasts into a pro-inflammatory phenotype.'

Minor comments:

- Fig. 4B: how many times was this experiment done/how many mice? How were the data quantified?

Images in figure 4B (original manuscript) are from one mouse only. The staining was done in at least 3 mice. Some of these are shown below, from 2 CIA mice with high inflammatory scores. Vimentin was included to identify stromal cells. Dotted lines delineate bone area, dotted boxes are areas of fibroblast proliferation. Images corroborate the results shown in figure 4b, as CIA synovium has a reduced SNA binding compared to MAA. We did not quantify the data, extensive quantitative data (transcriptomics and glycomics) are provided with isolated cells in other figures.

- the manuscript needs some proof reading

Thanks, this has been checked.

REVIEWERS' COMMENTS

Reviewer #1 (Remarks to the Author):

none

Reviewer #2 (Remarks to the Author):

Wang Y, et al., have performed a thorough adaptation of the manuscript and have answered my comments to satisfaction. The inclusion of experiments to prove the direct relation between sialylation and inflammation is a good addition to prove the causality of the story. The addition of O-glycosylation analysis is a very thorough as well, and the remaining uncertainties and caveats to the analyses have been handled much better in the text. I would find it helpful to still include Figure Q5-Reviewer 2 in the manuscript Supplemental Material, but find no reason to suggest further major revisions.

Reviewer #3 (Remarks to the Author):

The authors have provided an extensive revision of their manuscript in which my major concerns have been sufficiently addressed. In particular, the authors now provide a much more balanced and detailed distinction between the murine and the human data, and have added figure 9 which provides further functional evidence for the effect of reduced sialylation of cellular (surface) glycoproteins on SF function. Also, the detailed comparison between the murine and human SF glycome is an important addition, as are the clarifications with regard to figure 7.